# Evolutionary game model of intellectual property pledge financing between technology-based SMEs and banks based on the EVCC

**Li-na Dong**[1,2], **Mu Zhang**[1,2]*

1 School of Applied Economics, Guizhou University of Finance and Economics, Guiyang, China,
2 Guizhou Institution for Technology Innovation & Entrepreneurship Investment, Guizhou University of Finance and Economics, Guiyang, China

* rim_007@163.com

## Abstract

This study aims to improve the market efficiency of intellectual property pledge financing, based on the perspective of willingness to perform of technology-based SMEs, this paper defined the end-of-period value conversion coefficient of pledged property (EVCC) to measure the comparative relationship between the end-of-period value of the pledged intellectual property and the sum of principal and interest of the loan and introduced it into the game payment matrix; using evolutionary game theory, based on the assumption of bounded rationality, an evolutionary game model of intellectual property pledge financing between technology-based SMEs and banks based on the EVCC was constructed, and a numerical simulation was then conducted. The results of asymptotic stability analysis showed that when a certain condition is met, the strategy combination (performance, loan) is the evolutionary stability strategy (ESS). The numerical simulation showed that the EVCC has a positive impact on the speed of technology-based SMEs choosing the performance strategy, and there is a positive threshold effect (The threshold is 0.90). The initial value of pledged intellectual property has a negative impact on the speed of technology-based SMEs choosing the performance strategy, and there is a reverse threshold effect (The threshold is 1250), as well as the pledge rate of intellectual property (The threshold is 0.375). However, the loan interest rate has no significant impact on the strategic choice of technology-based SMEs. In addition, the EVCC has no significant impact on the banks' strategy choice. The initial value of pledged intellectual property has a negative impact on the speed of banks choosing loan strategies, and there is a reverse threshold effect (The threshold is 1250). The pledge rate of intellectual property has an inverted U-shaped impact on the speed of banks choosing loan strategies ($\omega^*$ may be close to 0.30), and there is a reverse threshold effect (The threshold is 0.375). The loan interest rate has a positive impact on the speed of banks choosing loan strategies, and there is a positive threshold effect (The threshold is 0.03). In

**Data availability statement:** This paper uses simulation data. All relevant data are presented in Table 4 of this paper.

**Funding:** This research was funded by the Research Project of Humanities and Social Sciences of Colleges and Universities in Guizhou, grant number 2024RW96, "Research on the problem of 'financing difficulties' and 'expensive financing' of small and medium-sized enterprises in Guizhou". The funders had no role in study design, data collection and analysis, decision to publish, or preparation of the manuscript.

**Competing interests:** The authors have declared that no competing interests exist.

addition, the trustworthy joint incentive not only has a positive impact on the speed of technology-based SMEs choosing the performance strategy, but also has a positive impact on the speed of banks choosing the loan strategy, and both have a positive threshold effect (The threshold for both is 15), as well as the dishonesty joint punishment (The threshold for both is 85). This model enriches the multi-agent game theory framework of intellectual property pledge financing. The numerical simulation results can provide a decision-making reference for technology-based SMEs and banks to formulate intellectual property pledge financing strategies.

## 1. Introduction

Intellectual property pledge financing is a financing method that the intellectual property suitable holder pledges the patent rights, registered trademark rights, copyright, and other intellectual property rights that are legally owned and are still valid, obtains funds from financial institutions such as banks, and repays the principal and interest of the funds on time ("Notice on Strengthening Intellectual Property Pledge Financing and Evaluation Management to Support the Development of Small and Medium-sized Enterprises (Finance and Enterprise No.199)"). In recent years, under the wave of data capitalization, the extension of intellectual property pledge financing has extended to the data field, and data intellectual property pledge financing has emerged. Data intellectual property pledge financing is a new type of financing method that uses data legally owned by enterprises and certified by a data intellectual property registration system or depository platform as a pledge, which is of positive significance for promoting the release of intrinsic value of data elements (https://www.cnipa.gov.cn/art/2023/9/25/art_53_187785.html). Technology-based SMEs refer to those small and medium-sized enterprises that have a certain number of scientific and technological personnel, possess independent intellectual property rights, proprietary technology or advanced knowledge, carry out innovative activities through scientific and technological investment, and provide products or services. The intellectual property rights (patents, registered trademarks, copyrights, data, etc.) owned by technology-based SMEs can provide vital information for banks to judge the future profitability and R&D level of enterprises. They can also effectively constrain enterprises' "moral hazard" and play a risk mitigation function after loans [1]. Intellectual property pledge financing provides a new financing way for asset-light technology-based SMEs, which helps technology-based SMEs alleviate financing constraints, reduce financing costs, and improve financing efficiency. Intellectual property pledge financing has become essential for technology-based SMEs to revitalize intangible assets, build market advantages, and enhance the kinetic energy of innovation and development.

However, due to many reasons, such as low commercial value, complex value assessment, difficult disposal and realization, information asymmetry, imperfect risk sharing, and compensation mechanisms, the development of intellectual property pledge financing business is restricted to a certain extent [1,2]. Market practice

shows that there may be multiple subjects, such as borrowers, lenders, third-party intermediaries (or platforms), and governments in intellectual property pledge financing. Analyzing the game relationship between various subjects is conducive to achieving market equilibrium and improving market efficiency.

In the intellectual property pledge financing business, there is a direct lending relationship between enterprises and banks. Therefore, enterprises and banks are the two most basic game subjects. At present, there are abundant research results on the game between enterprises and banks in academia [3]. Scholars mainly discuss the game between enterprises and banks from the perspective of loan business development [4–6], the game between enterprises and banks from the perspective of loan risk prevention and control [7–13], the game between enterprises and banks under the influence of government policies [14–16], the game between enterprises and banks under the influence of third-party intermediaries (or platforms) [9,17], the game between enterprises and banks under the background of big data [18], the game between enterprises and banks under the background of blockchain [19,20], the game between enterprises and banks under the background of financial technology [21–22], the game between enterprises and banks under the background of artificial intelligence [23], the signal game between enterprises and banks [21,24–27] and other issues. However, the above studies have not considered the impact of the comparative relationship between the end-of-period value of pledged intellectual property and the sum of principal and interest of loan on the willingness of enterprises to perform. Some existing studies have shown that in inventory pledge financing when the end-of-period value of inventory is lower than the sum of principal and interest of loan, the enterprises may default [28]. In view of this, this paper intends to define the end-of-period value conversion coefficient of pledged property (EVCC) to measure the comparative relationship between the end-of-period value of the pledged intellectual property and the sum of principal and interest of the loan based on the perspective of the enterprise's willingness to perform, and introduce it into the game payment matrix; using evolutionary game theory [29,30], based on the assumption of bounded rationality, construct an evolutionary game model of intellectual property pledge financing between technology-based SMEs and banks based on the EVCC; through numerical simulation, analyze the influence of the initial probability change of the strategic choice of technology-based SMEs and banks on the evolution path of the system, and probe into the influence of the change of the EVCC and other key parameters on the strategic choice of the two participants.

The remaining part of this paper is structured as follows: The Evolutionary Game Model section constructs an evolutionary game model of intellectual property pledge financing between technology-based SMEs and banks based on the EVCC; The Numerical Simulations section carries out the numerical simulation; The Discussions section discusses the results obtained and the Conclusions section concludes this paper.

## 2. Evolutionary game model

Based on the description of the evolutionary game problem, this section defines the end-of-period value conversion coefficient of pledged property (EVCC); puts forward the model hypothesis, and establishes the game payment matrix between technology-based SMEs and banks; constructs the replication dynamic equation and carries out the asymptotic stability analysis.

### 2.1 Description of evolutionary game problem

The own funds for technological innovation of technology-based SMEs cannot meet the demand for technological innovation investment. Therefore, the intellectual property, such as patent rights, registered trademark rights, copyrights data, and so on, that are legally owned by enterprises and still valid are pledged, enterprises apply to banks for intellectual property pledge loans. After the credit evaluation for technology-based SMEs, according to the evaluation value of intellectual property, according to a certain pledge rate, the bank issues a certain amount of pledge loan with a fixed loan interest rate and term to technology-based SMEs.

Due to the uncertainty of the market environment, enterprises apply to banks for intellectual property pledge loans, and banks may or may not lend; after the expiration of the pledge period, technology-based SMEs may perform or default;

therefore, the process of intellectual property pledge financing is a dynamic game process between technology-based SMEs and banks under bounded rationality.

## 2.2 Definition of EVCC

As we all know, two important factors affecting the debtor's credit are the ability to perform and the willingness to perform. The ability to fulfill the contract is an objective factor, and the willingness to fulfill the contract is a subjective factor. Both are indispensable.

The ability to perform mainly refers to the actual ability of the transaction subject to perform the economic contract. The ability to perform mainly includes the ability to pay and production capacity. The willingness to fulfill the contract generally refers to the ideas and thoughts of the subject of the transaction. The willingness to perform can be divided into active willingness to perform and passive willingness to perform; the active willingness to perform mainly depends on the personality and morality of the transaction subject, and the passive willingness to perform depends on the cost of default of the transaction subject (http://www.ndxj007.com/html/98761336.html). Previous studies have shown that the joint incentive for enterprises trustworthiness and the joint punishment for enterprises dishonesty are important factors affecting the willingness of enterprises to perform [12,17–19,22]. In addition, inspired by the literature [28], this paper believes that in intellectual property pledge financing, the comparative relationship between the end-of-period value of the pledged intellectual property and the sum of principal and interest of the loan is also an important factor affecting the willingness of enterprises to perform. Therefore, this paper defines the end-of-period value conversion coefficient of pledged property (EVCC) to measure the comparative relationship between the end-of-period value of the pledged intellectual property and the sum of the principal and interest of the loan.

Definition 1: Let $V_0$ be the initial value of the pledged intellectual property, $V_T$ be the end-of-period value of the pledged intellectual property, and $B$ be the sum of the principal and interest of the loan, $B = \omega V_0(1 + r_L)$, where $\omega$ is the pledge rate of intellectual property and $r_L$ is the loan interest rate. The end-of-period value conversion coefficient of pledged property (EVCC) is defined as:

$$k = \frac{V_T}{B} = \frac{V_T}{\omega V_0(1 + r_L)}$$

(1)

where $k$ represents EVCC, $k \geq 0$.

Obviously, when $k < 1$, we have $V_T < B$, the smaller $k$ is, the larger the degree of $V_T$ less than $B$ is, then the smaller the enterprises' willingness to perform (WP) is, the larger the enterprises' probability of default (PD) is, and vice versa; when $k \geq 1$, we have $V_T \geq B$, the larger the $k$ is, the larger the degree of $V_T$ greater than $B$ is, then the larger the enterprises' willingness to perform (WP) is, the smaller the enterprises' probability of default (PD) is, and vice versa. That means, $k$ is positively correlated with the enterprises' willingness to perform (WP), and $k$ is negatively correlated with the enterprises' probability of default (PD). The influence mechanism of $k$ on the enterprises' probability of default (PD) is: $k \uparrow \Rightarrow WP \uparrow \Rightarrow PD \downarrow$.

It can be seen from Equation (1) that when given the $V_T$, the smaller the $V_0$ is, the larger the $k$ is, then the smaller the enterprises' probability of default (PD) is, and vice versa. The smaller the $\omega$ is, the larger the $k$ is, then the smaller the enterprises' probability of default (PD) is, and vice versa. The smaller the $r_L$ is, the larger the $k$ is, then the smaller the enterprises' probability of default (PD) is, and vice versa.

## 2.3 Model hypotheses

**Hypothesis 1:** There are only two groups in the game: technology-based SMEs and banks. The participants in the game are independent decision-making individuals with limited rationality and limited information, that is, according to their

cognition and environment, they dynamically adjust their decisions in the process of playing games with each other, to maximize their expected returns. The strategy selection set of technology-based SMEs is $S_1$ = {performance, default}; the bank's strategy choice set is $S_2$ = {loan, no loan}.

**Hypothesis 2:** Let us assume that the probability of technology-based SMEs choosing the "performance" strategy is $x$ $(0 \leq x \leq 1)$, and the probability of technology-based SMEs choosing the "default" strategy is 1-$x$; the probability of banks choosing the "loan" strategy is $y$ $(0 \leq y \leq 1)$, and the probability of banks choosing the "no loan" strategy is 1-$y$.

**Hypothesis 3:** The own funds for technological innovation of technology-based SMEs is $M_0$, which cannot meet the investment demand of R&D projects, and the enterprises need to apply to the banks for pledge loans. Let us assume that the enterprises will apply for the pledge loans to the banks as the pledge of the intellectual property with the initial value of $V_0$ and will pay the evaluation fee $C_P$ of the pledge intellectual property; Let us assume that the intellectual property pledge rate given by the banks is $\omega$, the amount of intellectual property pledge loans obtained by the enterprises is $\omega V_0$, and the sum of principal and interest of the loans is $B = \omega V_0(1 + r_L)$, where, $r_L$ is the loan interest rate.

**Hypothesis 4:** Technology-based SMEs put their own funds $M_0$ for technological innovation and intellectual property pledge loans $\omega V_0$ together into R&D projects. The rate of return of enterprises investment in R&D projects is $r_I$, $r_I$ is the main factor affecting the performance ability of technology-based SMEs. The larger the $r_I$ is, the stronger the enterprises' ability to perform (PC), the smaller the enterprises' probability of default (PD), and vice versa. For simplicity, it may be assumed that: when $r_I > 0$, the enterprises investment in R&D projects is successful; when $r_I \leq 0$, the enterprises investment in R&D projects is a failure. Let us assume that $r_I > 0$ is a necessary condition for enterprises performance, that is, the occurrence of enterprises performance must depend on $r_I > 0$, but when $r_I > 0$ occurs, enterprises performance does not necessarily occur; $r_I \leq 0$ is a necessary condition for enterprises default, that is, the occurrence of enterprises default must depend on $r_I \leq 0$, but when $r_I \leq 0$ occurs, enterprises default does not necessarily occur.

Without loss of generality, when the enterprises adopt the performance strategy, let us assume that the R&D projects investment return rate of the enterprises is $r_{Ih}$ ($r_{Ih} > 0$), and the R&D projects investment return of the enterprises is $r_{Ih}(M_0 + \omega V_0)$; when the enterprises adopt the default strategy, let us assume that the R&D projects investment return rate of the enterprises is $-r_{Il}$ ($-r_{Il} \leq 0$, $r_{Il} \geq 0$), and the R&D project investment return of the enterprises is $-r_{Il}(M_0 + \omega V_0)$.

**Hypothesis 5:** Let the end-of-period value of the pledged intellectual property be $V_T$, $V_T = kB$, where $k$ ($k \geq 0$) is the EVCC. $k$ is an important factor affecting the willingness of technology-based SMEs to perform. Let $V_T \geq B$ ($k \geq 1$) be a necessary condition for enterprises performance, that is, the occurrence of enterprises performance must depend on $k \geq 1$, but when $k \geq 1$ occurs, enterprises performance does not necessarily occur; $V_T < B$ ($k < 1$) is a necessary condition for enterprises default, that is, the occurrence of enterprises default must depend on $k < 1$, but when $k < 1$ occurs, enterprises default does not necessarily occur. Let us assume that the joint incentive for technology-based SMEs trustworthiness is $E$, and the joint punishment for technology-based SMEs dishonesty is $F$. In addition, when the banks adopt the no loan strategy, the expected return of technology-based SMEs is $-C_P$.

**Hypothesis 6:** When the banks adopt the loan strategy, the intellectual property pledge rate given by the banks is $\omega$, $\omega V_0$ is the principal of the bank's intellectual property pledge loans; Let us assume that the loan interest rate is $r_L$, the deposit interest rate is $r_D$. Banks issuing intellectual property pledge loans to technology-based SMEs will pay credit evaluation costs $C_C$ and post-supervision costs $C_S$. After the expiration of the pledge period, if the technology-based SMEs perform, the banks will obtain interest income $\omega V_0 r_L$; if the technology-based SMEs default, the banks will lose the principal and interest of the loans. Currently, the banks will auction and sell the pledged intellectual property according to the law, and the disposal costs are $C_D$. In addition, when the banks adopt the no loan strategy, the expected return of the banks is $-C_C$.

The symbols and meanings of the relevant variables are shown in Table 1.

**Table 1. Symbols and meanings of related variables.**

| Participants | Symbols | Meanings |
|---|---|---|
| Technology-based SMEs | $C_P$ | Pledged intellectual property evaluation fee. |
| | $r_I$ | The rate of return of enterprise investment in R&D projects. |
| | $r_{Ih}$ | The rate of return when the enterprise invests in the success of the R&D project. |
| | $-r_{Il}$ | The rate of return when the enterprise invests in R&D project failure. |
| | $M_0$ | The own funds for technological innovation of technology-based SMEs. |
| | $V_0$ | Initial value of pledged intellectual property. |
| | $V_T$ | The end-of-period value of pledged intellectual property. |
| | $k$ | EVCC. |
| | $E$ | The joint incentive for enterprises trustworthy. |
| | $F$ | The joint punishment for enterprises dishonesty. |
| Banks | $C_C$ | The credit evaluation costs of banks. |
| | $C_S$ | Post-supervision costs of banks. |
| | $C_D$ | The disposal costs of pledge intellectual property. |
| | $\omega$ | Intellectual property pledge rate. |
| | $r_L$ | The loan interest rate. |
| | $r_D$ | The deposit interest rate. |
| | $B$ | The sum of principal and interest of the loans. |

## 2.4 Game payoff matrix

When technology-based SMEs and banks carry out evolutionary games, the expected returns of game participants are different under different strategy combinations.

If the technology-based SMEs adopt the performance strategy and the banks adopt the loan strategy, that is, (performance, loan), the expected return of the technology-based SMEs is: $-C_P + \omega V_0 + r_{Ih}(M_0 + \omega V_0) - \omega V_0(1 + r_L) + E$; the banks' expected return is: $-C_C - C_S + \omega V_0(r_L - r_D)$.

If the technology-based SMEs adopt the performance strategy and the banks adopt the no loan strategy, that is, (performance, no loan), the expected return of the technology-based SMEs is: $-C_P$; the banks' expected return is: $-C_C$.

If the technology-based SMEs adopt the default strategy and the banks adopt the loan strategy, that is, (default, loan), the expected return of the technology-based SMEs is: $-C_P + \omega V_0 - r_{Il}(M_0 + \omega V_0) - kB - F$; the banks' expected return is: $-C_C - C_S - C_D + kB - \omega V_0(1 + r_L) - \omega V_0 r_D$. According to Hypothesis 5, there is $k < 1$.

If the technology-based SMEs adopt the default strategy and the banks adopt the no loan strategy, that is, (default, no loan), the expected return of the technology-based SMEs is: $-C_P$; the banks' expected return is: $-C_C$.

In summary, the game payment matrix of technology-based SMEs and banks is shown in Table 2.

## 2.5 Replication dynamic equation

According to Table 2, the expected return of technology-based SMEs adopting the performance strategy is:

$$U_{e,1} = y\left(-C_P + \omega V_0 + r_{Ih}(M_0 + \omega V_0) - \omega V_0(1 + r_L) + E\right) + (1 - y)\left(-C_P\right) \tag{2}$$

The expected return of technology-based SMEs adopting the default strategy is:

$$U_{e,2} = y\left(-C_P + \omega V_0 - r_{Il}(M_0 + \omega V_0) - kB - F\right) + (1 - y)\left(-C_P\right) \tag{3}$$

**Table 2. Game payoff matrix.**

| Expected return under different strategy combinations | | Banks | |
|---|---|---|---|
| | | **Loan (*y*)** | **No Loan (1-*y*)** |
| Technology-based SMEs | Performance (*x*) | $-C_P + \omega V_0 + r_{lh}(M_0 + \omega V_0) - \omega V_0(1 + r_L) + E$ <br> $-C_C - C_S + \omega V_0(r_L - r_D)$ | $-C_P$ <br> $-C_C$ |
| | Default (1-*x*) | $-C_P + \omega V_0 - r_{ll}(M_0 + \omega V_0) - kB - F$ <br> $-C_C - C_S - C_D + kB - \omega V_0(1 + r_L) - \omega V_0 r_D$ | $-C_P$ <br> $-C_C$ |

Note: In parentheses () is probability.

The average expected return of the performance and default strategies selected by the technology-based SMEs with the probability of *x* and 1-*x* respectively is:

$$\overline{U_e} = xU_{e,1} + (1-x)U_{e,2} \tag{4}$$

The expected return of the banks adopting the loan strategy is:

$$U_{b,1} = x(-C_C - C_S + \omega V_0(r_L - r_D)) + (1-x)(-C_C - C_S - C_D + kB - \omega V_0(1 + r_L) - \omega V_0 r_D) \tag{5}$$

The expected return of the banks adopting the no loan strategy is:

$$U_{b,2} = x(-C_C) + (1-x)(-C_C) = -C_C \tag{6}$$

The average expected return of the loan and no loan strategies selected by the banks with the probability of *y* and 1-*y* respectively is:

$$\overline{U_b} = yU_{b,1} + (1-y)U_{b,2} \tag{7}$$

According to the Malthusian Dynamic Equation [29], the growth rate (*dx/dt*) of technology-based SMEs adopting the performance strategy is equal to the difference between the expected return ($U_{e,1}$) and the average expected return ($\overline{U_e}$), multiplied by *x*; the growth rate (*dy/dt*) of the banks adopting the loan strategy is equal to the difference between the expected return ($U_{b,1}$) and the average expected return ($\overline{U_b}$), multiplied by *y*. Therefore, the replication dynamic equation of technology-based SMEs and banks respectively is:

$$\frac{dx}{dt} = x(U_{e,1} - \overline{U_e}) = x(1-x)(U_{e,1} - U_{e,2})$$
$$= x(1-x)y((r_{lh} + r_{ll})(M_0 + \omega V_0) - \omega V_0(1 + r_L) + kB + E + F) \tag{8}$$

$$\frac{dy}{dt} = y(U_{b,1} - \overline{U_b}) = y(1-y)(U_{b,1} - U_{b,2})$$
$$= y(1-y)(x(-C_S + \omega V_0(r_L - r_D)) + (1-x)(-C_S - C_D + kB - \omega V_0(1 + r_L) - \omega V_0 r_D))$$
$$= y(1-y)[x(\omega V_0 + 2\omega V_0 r_L + C_D - kB) + (-C_S - C_D + kB - \omega V_0(1 + r_L) - \omega V_0 r_D)] \tag{9}$$

## 2.6 Asymptotic stability analysis

Let $dx/dt = 0$, $dy/dt = 0$, the local equilibrium points of the replication dynamic system can be obtained. In the asymmetric game, the asymptotic stability point (i.e., evolutionary stable strategy, ESS) of the replicator dynamic system of

multi-group evolutionary game is a strict Nash equilibrium., that is, the pure strategy equilibrium [31]. Therefore, the mixed strategy equilibrium must not be an evolutionarily stable strategy [31]. Therefore, for the evolutionary game between technology-based SMEs and banks in intellectual property pledge financing, only pure strategies are considered, and four local equilibrium points of the replication dynamic system are obtained: $E_1 = (0,0)$, $E_2 = (0,1)$, $E_3 = (1,0)$, $E_4 = (1,1)$.

According to the Friedman method [32], the asymptotic stability of the local equilibrium point can be judged by the signs of determinant $Det(J)$ and trace $Tr(J)$ of the Jacobian matrix of the replicator dynamic system. When $Det(J) > 0$ and $Tr(J) < 0$, the corresponding equilibrium point is the asymptotically stable point (i.e., evolutionarily stable strategy, ESS); when $Det(J) > 0$ and $Tr(J) > 0$, the corresponding equilibrium point is unstable point; when the signs of $Det(J)$ and $Tr(J)$ appears in other cases, the corresponding equilibrium point is the saddle point.

By solving the partial derivatives of $x$ and $y$ in (8) and (9) respectively, the Jacobian matrix $J$ of the replication dynamic system is obtained as:

$$J = \begin{bmatrix} (1-2x)\, yQ_1 & x\,(1-x)\, Q_1 \\ y\,(1-y)\, Q_2 & (1-2y)\,(xQ_2 + Q_3) \end{bmatrix}$$

(10)

where

$$Q_1 = (r_{Ih} + r_{II})\,(M_0 + \omega V_0) - \omega V_0\,(1 + r_L) + kB + E + F$$

$$Q_2 = 2\omega V_0 r_L + C_D - kB + \omega V_0$$

$$Q_3 = -C_S - C_D + kB - \omega V_0\,(1 + r_L) - \omega V_0 r_D$$

From [Equation (10)](), the determinant $Det(J)$ and trace $Tr(J)$ of the Jacobian matrix is respectively obtained as:

$$Det(J) = (1-2x)\,(1-2y)\, yQ_1\,(xQ_2 + Q_3) - xy\,(1-x)\,(1-y)\, Q_1 Q_2$$

(11)

$$Tr(J) = (1-2x)\, yQ_1 + (1-2y)\,(xQ_2 + Q_3)$$

(12)

By substituting the four local equilibrium points $E_1, E_2, E_3, E_4$ into [Equations (11)]() and [(12)]() respectively, the determinant $Det(J)$ and trace $Tr(J)$ of the corresponding Jacobian matrix are obtained, and the results are shown in [Table 3](). According to the signs of determinant $Det(J)$ and trace $Tr(J)$ of the Jacobian matrix of the local equilibrium point, the Friedman method [32] is used to judge the asymptotic stability of the local equilibrium point. The results are shown in [Table 3]().

It can be seen from [Table 3]() that the determinant of the Jacobian matrix corresponding to the local equilibrium point $E_1 = (0,0)$ is 0, so the local equilibrium point $E_1 = (0,0)$ is a saddle point. The trace of the Jacobian matrix corresponding to the local equilibrium point $E_2 = (0,1)$ is greater than 0, so the local equilibrium point $E_2 = (0,1)$ is an unstable point or saddle point. The determinant of the Jacobian matrix corresponding to the local equilibrium point $E_3 = (1,0)$ is 0, so the local equilibrium point $E_3 = (1,0)$ is a saddle point. When the condition ① is satisfied, the local equilibrium point $E_4 = (1,1)$ is the asymptotically stable point (i.e., the evolutionarily stable strategy, ESS). If and only if $Q_1 > 0$, and when $Q_2 + Q_3 > 0$, $-Q_1 - (Q_2 + Q_3) < 0$, the condition ① holds, the evolutionary stability strategy is then analyzed as follows:

When the condition ① is satisfied, the local equilibrium point $E_4 = (1,1)$ is the asymptotic stable point of the replication dynamic system, that is, the strategy combination (performance, loan) is the evolutionary stable strategy (ESS).

**Table 3. Asymptotic stability analysis of local equilibrium points.**

| Point of Equilibrium ($x,y$) | Det(J) | Tr(J) | sign | Stability |
|---|---|---|---|---|
| (0,0) | 0 | $Q_3$ | (0, N) | Saddle point |
| (0,1) | $-Q_1 Q_3$ | $Q_1 - Q_3$ | (N, +) | Unstable point or saddle point |
| (1,0) | 0 | $Q_2 + Q_3$ | (0, N) | Saddle point |
| (1,1) | $Q_1 (Q_2 + Q_3)$ | $-Q_1 - (Q_2 + Q_3)$ | (N, N) | ESS (when the condition ① is met) |

Note: 1) "+" denotes a positive value, "-" denotes a negative value, and "N" denotes uncertainty.

2) Condition ①: $Q_1 (Q_2 + Q_3) > 0$, $-Q_1 - (Q_2 + Q_3) < 0$.

When $Q_1 > 0$, and when $Q_2 + Q_3 > 0$, there is $(r_{Ih} + r_{Il})(M_0 + \omega V_0) + kB + E + F > \omega V_0 (1 + r_L)$, and $\omega V_0 r_L > C_S + \omega V_0 r_D$. On the one hand, for technology-based SMEs, when the sum of the return on the success of the enterprises investment R&D projects, the absolute value of the return on the failure of the enterprises investment R&D projects, the end-of-period value of the pledged intellectual property, the trustworthy joint incentive and the dishonesty joint punishment for enterprises are greater than the sum of the principal and interest of the intellectual property pledge loan, the technology-based SMEs will adopt the performance strategy. On the other hand, for banks, when the loan interest income is greater than the sum of the post-supervision cost and the deposit interest expenditure, the banks will adopt the loan strategy.

## 3. Numerical simulations

Based on parameter assignment, this section analyzes the influence of the initial probability change of the strategic choice of the technology-based SMEs and banks on the evolution path of the system and the influence of the change of six key parameters on the strategic choice of the two participants through numerical simulation.

### 3.1 Parameter assignment

According to the results of asymptotic stability analysis, the parameter condition that satisfies the Pareto optimal state (1,1) is condition ①. Based on satisfaction $(r_{Ih} + r_{Il})(M_0 + \omega V_0) + V_T + E + F > \omega V_0 (1 + r_L)$ and $\omega V_0 r_L > C_S + \omega V_0 r_D$, according to the actual operation of China's intellectual property pledge financing business, combined with relevant literatures, the following parameters are set as follows:

The R&D project investment budget of technology-based SMEs is set at 5 million yuan; the own funds $M_0$ for technological innovation of technology-based SMEs is 2 million yuan (CNY); the pledge rate $\omega$ of intellectual property given by the banks is 0.30 (Reference source: "Operating Procedures for Patent Pledge Loans of Tianjin Rural Commercial Bank"); the initial value $V_0$ of pledged intellectual property is 10 million yuan (In this way, the enterprises will receive a pledged loan of 3 million yuan.); the EVCC $k$ is 0.90 (Reference source: [14]); the trustworthy joint incentive $E$ for the technology-based SMEs is 300,000 yuan (Reference source: [13]), and the dishonesty joint punishment $F$ for the technology-based SMEs is 1 million yuan (Reference source: [13]); the rate of return $r_{Ih}$ when the enterprises invest in the success of R&D projects is 0.10 (Reference source: [33]); the rate of return $-r_{Il}$ when the enterprises invest in R&D projects failure is −0.10 (Reference source: [33]). In addition, the banks' post-supervision cost $C_S$ is 10,000 yuan (Reference source: General charging standards for post loan management fees in the Chinese banking industry); the disposal cost of pledged intellectual property $C_D$ is 200,000 yuan (Reference source: General charging standards for intellectual property assessment fees, auction or sale fees, legal fees, registration fees, other fees, etc. in China); the loan interest rate $r_L$ is 0.0365 (Reference source: Loan market quoted interest rate of the People's Bank of China (one-year term)), and the deposit interest rate $r_D$ is 0.017 (Reference source: Benchmark interest rate of fixed deposits of the People's Bank of China (one-year term)). The assignment of 12 parameters is shown in Table 4.

**Table 4. Parameter assignment.**

| Parameters | $M_0$ | $V_0$ | $k$ | $E$ | $F$ | $r_{lh}$ |
|---|---|---|---|---|---|---|
| Numerical Value | 200 | 1000 | 0.90 | 30 | 100 | 0.10 |
| Parameters | $-r_{ll}$ | $C_S$ | $C_D$ | $\omega$ | $r_L$ | $r_D$ |
| Numerical Value | −0.10 | 1 | 20 | 0.30 | 0.0365 | 0.017 |

## 3.2 Numerical simulation results

According to Table 4, the numerical simulation is carried out by using MATLAB software, and the influence of the initial probability change of the strategic choice of the technology-based SMEs and banks on the evolution path of the system is analyzed in detail, as well as the influence of the change of key parameters such as $k$, $V_0$, $\omega$, $r_L$, $E$ and $F$ on the strategic choice of two participants.

**3.2.1 The influence of the change of initial probability of two participants' strategy selection on the evolution path of the system.** Assuming that the initial probability of the technology-based SMEs to choose the performance strategy is $x_0$, and the initial probability of the banks to choose the loan strategy is $y_0$, let $(x_0, y_0)$ be (0.200,0.200), (0.250,0.250), (0.275,0.275), (0.300,0.300), (0.350,0.350), (0.400,0.400) and (0.450,0.450) respectively. According to Table 4, the influence of the initial probability changes of the strategic choice of the two participants on the evolution path of the system is obtained, as shown in Fig 1. The horizontal axis in Fig 1 indicates the probability $x$ of technology-based SMEs choosing the performance strategy, and the vertical axis indicates the probability $y$ of banks choosing the loan strategy. The curve in Fig 1 shows the evolution path of the system under different $(x_0, y_0)$ values. It can be seen from Fig 1, with the increase of $x_0, y_0$, the convergence speed of the system tends to Pareto optimal state is accelerated. And when $x_0, y_0 < 0.275$, the system does not converge to the Pareto optimal state (1,1); when $x_0, y_0 \geq 0.275$, the system converges to the Pareto optimal state (1,1). This shows that the higher the initial probability of the two participants' strategy selection

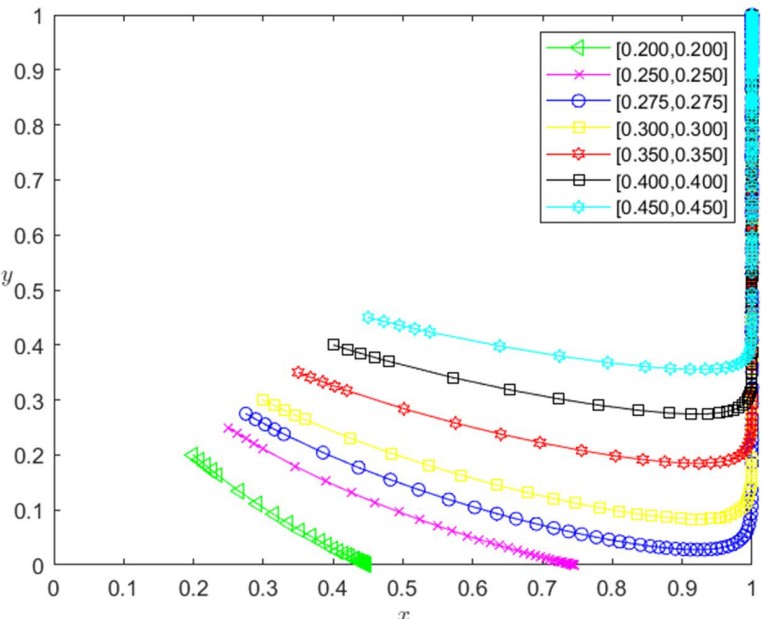

**Fig 1. The influence of the change of the initial probability of two participants' strategy selection on the evolution path of the system.**

is, the faster the speed of reaching the Pareto optimal state is; and the initial probability of the two participants' strategy selection needs to reach 27.5%, the system can achieve the Pareto optimal state (performance, loan).

### 3.2.2 The impact of the change of key parameters on the strategic choice of two participants.

(1) The impact of $k$ change on the strategic choice of two participants.

Let the initial probability $x_0 = 0.275$, $y_0 = 0.275$, let $k$ be 0.75, 0.80, 0.85, 0.90, 0.93, 0.96 and 0.99, respectively. Under the condition that other parameters remain unchanged, the influence of the change of $k$ on the evolution path of the system is shown in Fig 2. The horizontal axis in Fig 2 indicates the probability $x$ of technology-based SMEs choosing the performance strategy, and the vertical axis indicates the probability $y$ of banks choosing the loan strategy. The curve in Fig 2 shows the evolution path of the system under different $k$ values. It can be seen from Fig 2 that with the increase of $k$, the convergence speed of the system tends to Pareto optimal state is accelerated. When $k < 0.90$, the system does not converge to the Pareto optimal state (1,1). When $k \geq 0.90$, the system converges to the Pareto optimal state (1,1).

Fig 3 reflects the impact of the change of $k$ on the strategic choice of technology-based SMEs. The horizontal axis in Fig 3 represents the number $t$ of simulation steps, and the vertical axis indicates the probability $x$ of technology-based SMEs choosing the performance strategy. The curve in Fig 3 shows the evolution path of $x$ under different $k$ values. It can be seen from Fig 3 that the larger $k$ is, the faster the convergence rate of $x$ approaching 1 is. And when $k < 0.90$, $x$ converges below 0.7; when $k \geq 0.90$, $x$ converges to 1. This shows that the change of $k$ has a positive impact on the evolution path of $x$. The larger the $k$ is, the faster the technology-based SMEs choosing the performance strategy. And the influence of $k$ change on the evolution path of $x$ has a positive threshold effect. After $k$ is higher than a certain threshold, the technology-based SMEs will choose the performance strategy. In addition, the numerical simulation results show that the change of $k$ has no significant impact on the banks' strategy choice.

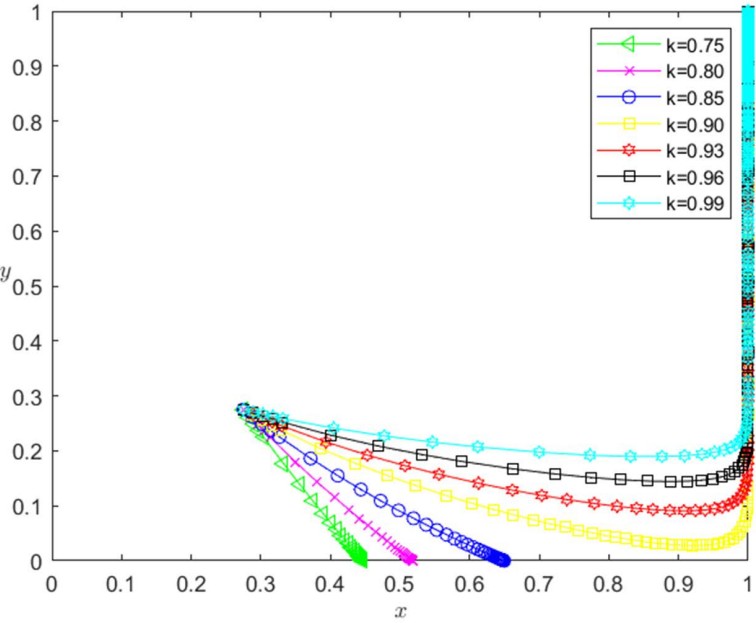

**Fig 2. The influence of the change of $k$ on the evolution path of the system.**

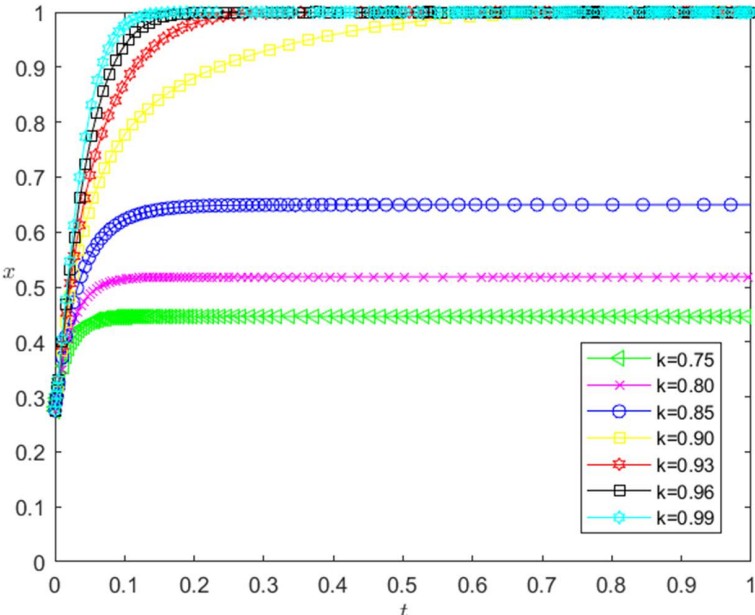

**Fig 3. The impact of the change of *k* on the technology-based SMEs' strategy choice.**

(2) The impact of $V_0$ change on the strategic choice of two participants.

Let the initial probability $x_0 = 0.275$, $y_0 = 0.275$, and the initial value of the pledged intellectual property be 900, 1000, 1100, 1200, 1250, 1350 and 1500, respectively. When other parameters remain unchanged, the impact of the initial value change of the pledged intellectual property on the system evolution path is shown in Fig 4. The horizontal axis in Fig 4 indicates the probability *x* of technology-based SMEs choosing the performance strategy, and the vertical axis indicates the probability *y* of banks choosing the loan strategy. The curve in Fig 4 shows the evolution path of the system under different $V_0$ values. It can be seen from Fig 4 that with the decrease of $V_0$, the convergence speed of the system tends to Pareto optimal state is accelerated. And when $V_0 > 1250$, the system does not converge to the Pareto optimal state (1,1); when $V_0 \leq 1250$, the system converges to the Pareto optimal state (1,1).

Fig 5 reflects the impact of the initial value $V_0$ change of pledged intellectual property on the strategic choice of technology-based SMEs. The horizontal axis in Fig 5 represents the number *t* of simulation steps, and the vertical axis indicates the probability *x* of technology-based SMEs choosing the performance strategy. The curve in Fig 5 shows the evolution path of *x* under different $V_0$ values. It can be seen from Fig 5 that the smaller $V_0$ is, the faster the convergence rate of *x* approaching 1 is. And when $V_0 > 1250$, *x* converges below 0.9; when $V_0 \leq 1250$, *x* converges to 1. This shows that the change of $V_0$ has a negative impact on the evolution path of *x*. The smaller the $V_0$ is, the faster the technology-based SMEs choosing the performance strategy is. And the influence of $V_0$ change on the evolution path of *x* has a reverse threshold effect. After $V_0$ is lower than a certain threshold, the technology-based SMEs will choose the performance strategy.

Fig 6 reflects the impact of $V_0$ change in the initial value of pledged intellectual property on banks' strategy choice. The horizontal axis in Fig 6 represents the number *t* of simulation steps, and the vertical axis indicates the probability *y* of banks choosing the loan strategy. The curve in Fig 6 shows the evolution path of *y* under different $V_0$ values. It can be seen from Fig 6 that the smaller $V_0$ is, the faster the convergence rate of *y* approaching 1 is. And when $V_0 > 1250$, *y* converges to 0; when $V_0 \leq 1250$, *y* converges to 1. This shows that the change of $V_0$ has a negative impact on the evolution path of *y*. The

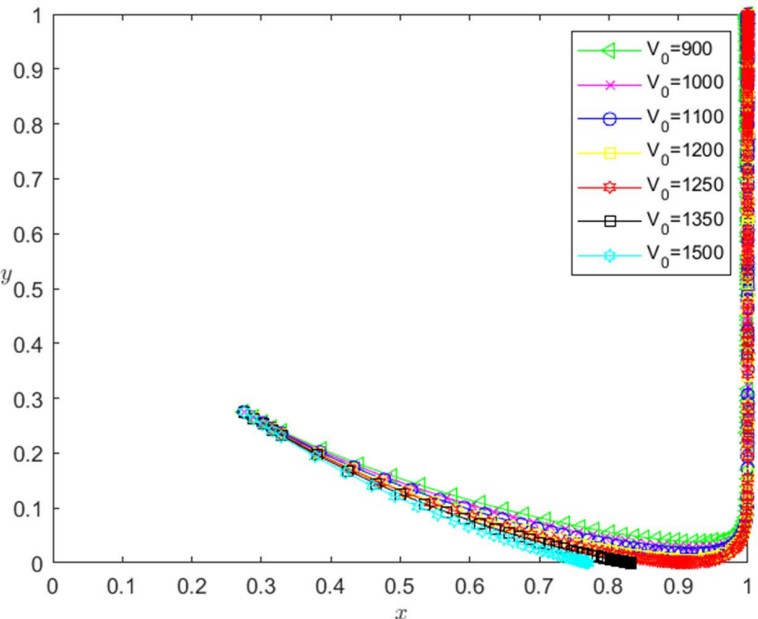

**Fig 4. The influence of the change of $V_0$ on the evolution path of the system.**

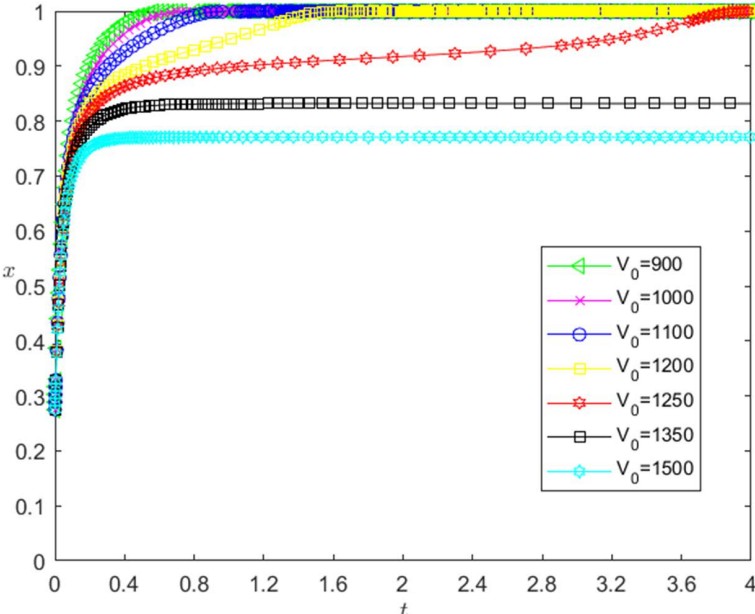

**Fig 5. The impact of the change of $V_0$ on the technology-based SMEs' strategic choice.**

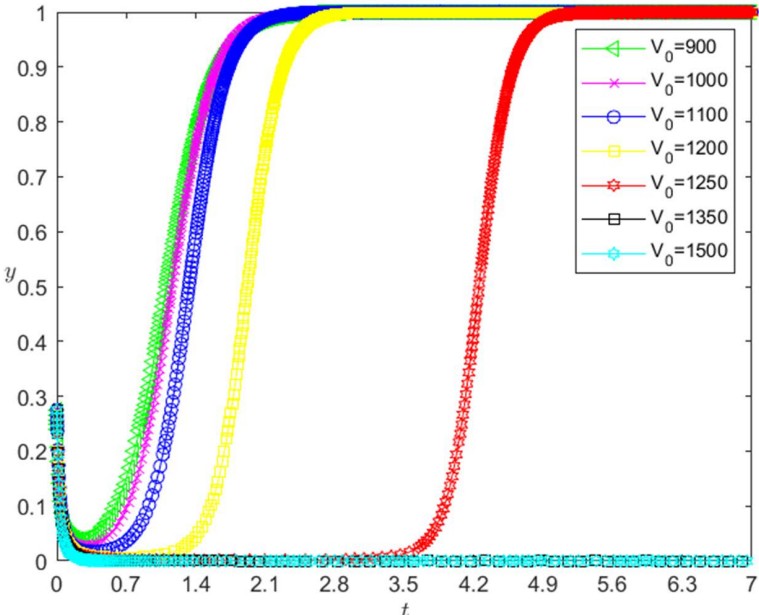

**Fig 6. The impact of the change of $V_0$ on the banks' strategy choice.**

smaller the $V_0$ is, the faster the banks choosing the loan strategy. And there is a reverse threshold effect on the influence of $V_0$ change on the evolution path of $y$. After $V_0$ is lower than a certain threshold, banks will choose the loan strategy.

(3) The impact of $\omega$ change on the strategic choice of two participants.

Let the initial probability $x_0 = 0.275$, $y_0 = 0.275$, the intellectual property pledge rate $\omega$ take 0.075, 0.150, 0.225, 0.300, 0.375, 0.450 and 0.525, respectively. When other parameters remain unchanged, the influence of $\omega$ change on the evolution path of the system is shown in Fig 7. The horizontal axis in Fig 7 indicates the probability $x$ of technology-based SMEs choosing the performance strategy, and the vertical axis indicates the probability $y$ of banks choosing the loan strategy. The curve in Fig 7 shows the evolution path of the system under different $\omega$ values. It can be seen from Fig 7 that with the decrease of $\omega$, the convergence speed of the system tends to the Pareto optimal state is accelerated. When $\omega > 0.375$, the system does not converge to the Pareto optimal state (1,1); when $\omega \leq 0.375$, the system converges to the Pareto optimal state (1,1).

Fig 8 reflects the impact of $\omega$ change on the strategic choice of technology-based SMEs. The horizontal axis in Fig 8 represents the number $t$ of simulation steps, and the vertical axis indicates the probability $x$ of technology-based SMEs choosing the performance strategy. The curve in Fig 8 shows the evolution path of $x$ under different $\omega$ values. It can be seen from Fig 8 that the smaller $\omega$ is, the faster the convergence rate of $x$ approaching 1 is. And when $\omega > 0.375$, $x$ converges below 0.8; when $\omega \leq 0.375$, $x$ converges to 1. This shows that the change of $\omega$ has a negative impact on the evolution path of $x$. The smaller $\omega$ is, the faster the technology-based SMEs choosing the performance strategy. And the influence of $\omega$ change on the evolution path of $x$ has a reverse threshold effect. When $\omega$ is lower than a certain threshold, the technology-based SMEs will choose the performance strategy.

Fig 9 reflects the impact of $\omega$ change on banks' strategy choice. The horizontal axis in Fig 9 represents the number $t$ of simulation steps, and the vertical axis indicates the probability $y$ of banks choosing the loan strategy. The curve in Fig 9 shows the evolution path of $y$ under different $\omega$ values. It can be seen from Fig 9 that when $\omega < 0.30$, the greater the $\omega$ is, the faster the convergence rate of $y$ approaching 1 is; when $\omega \geq 0.30$, the larger $\omega$ is, the slower the convergence rate of $y$ approaching 1 is. And when $\omega > 0.375$, $y$ converges to 0; when $\omega \leq 0.375$, $y$ converges to 1. This shows that the $\omega$ change

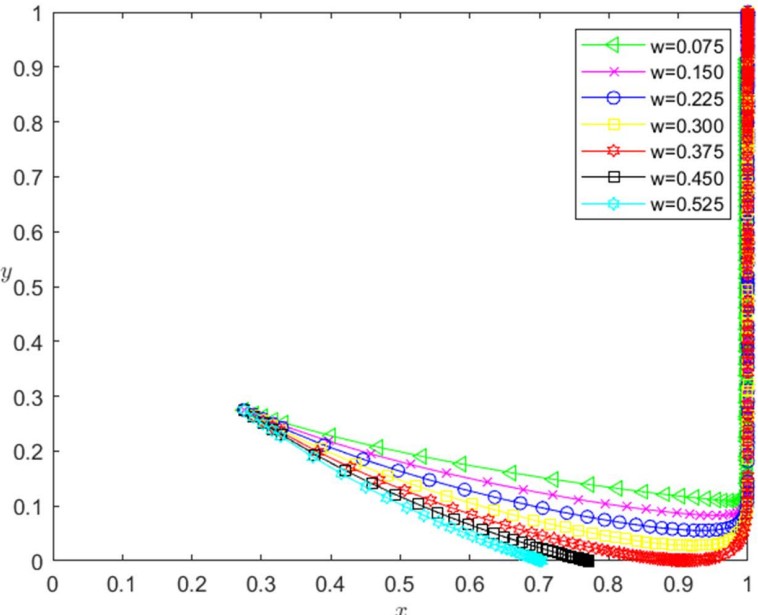

**Fig 7. The influence of the change of *ω* on the evolution path of the system.**

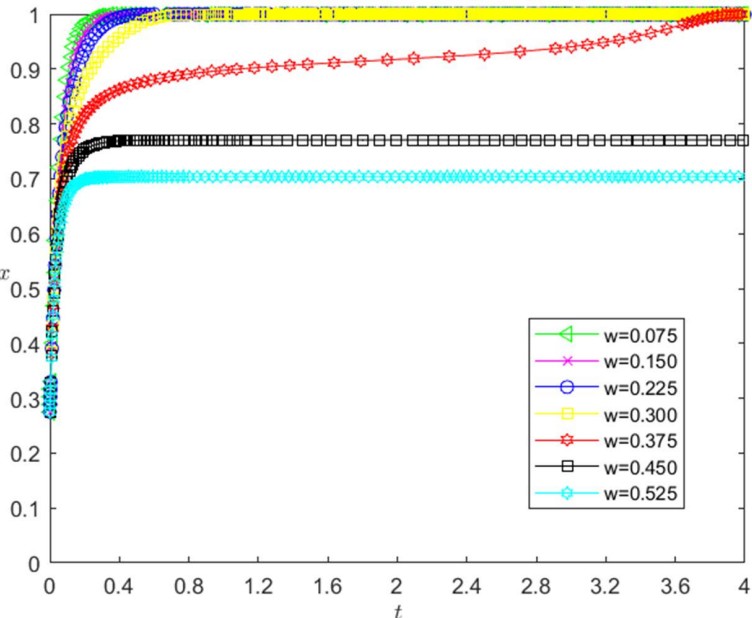

**Fig 8. The impact of the change of *ω* on the technology-based SMEs' strategic choice.**

has an inverted U-shaped effect on the evolution path of $y$, and there may be a $\omega^*$ that makes the banks choose the loan strategy the fastest. When $\omega < \omega^*$, the greater the $\omega$ is, the faster the banks choosing the loan strategy is; when $\omega > \omega^*$, the greater the $\omega$ is, the slower the banks choosing the loan strategy is. There is a reverse threshold effect on the influence of $\omega$ change on the evolution path of $y$. When $\omega$ is lower than a certain threshold, banks will choose the loan strategy.

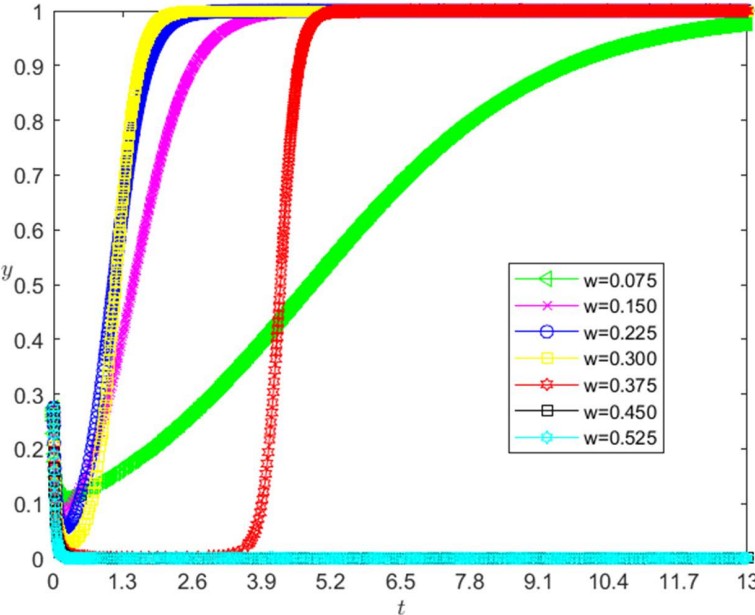

**Fig 9. The impact of the change of ω on the banks' strategy choice.**

(4)  The impact of $r_L$ change on the strategic choice of two participants.

Let the initial probability $x_0 = 0.275$, $y_0 = 0.275$, and the bank loan interest rate is taken as 0.0100, 0.0200, 0.0300, 0.0365, 0.0465, 0.0565 and 0.0665, respectively. When other parameters remain unchanged, the impact of bank loan interest rate change on the evolution path of the system is shown in Fig 10. The horizontal axis in Fig 10 indicates the probability $x$ of technology-based SMEs choosing the performance strategy, and the vertical axis indicates the probability $y$ of banks choosing the loan strategy. The curve in Fig 10 shows the evolution path of the system under different $r_L$ values. It can be seen from Fig 10 that with the increase of $r_L$, the convergence speed of the system tends to Pareto optimal state is accelerated. And when $r_L < 0.03$, the system does not converge to the Pareto optimal state (1,1); when $r_L \geq 0.03$, the system converges to the Pareto optimal state (1,1).

Fig 11 reflects the impact of $r_L$ change on banks' strategy choice. The horizontal axis in Fig 11 represents the number $t$ of simulation steps, and the vertical axis indicates the probability $y$ of banks choosing the loan strategy. The curve in Fig 11 shows the evolution path of $y$ under different $r_L$ values. It can be seen from Fig 11 that the larger the $r_L$ is, the faster the convergence rate of $y$ approaching 1 is. And when $r_L < 0.03$, $y$ converges to 0; when $r_L \geq 0.03$, $y$ converges to 1. This shows that the change of $r_L$ has a positive impact on the evolution path of $y$. The higher the $r_L$ is, the faster the banks choosing the loan strategy is. And the influence of $r_L$ change on the evolution path of $y$ has a positive threshold effect. After $r_L$ is higher than a certain threshold, the banks will choose the loan strategy. In addition, the numerical simulation results show that the change of $r_L$ has no significant effect on the strategic choice of technology-based SMEs.

(5)  The impact of $E$ change on the strategic choice of two participants.

Let the initial probability $x_0 = 0.275$, $y_0 = 0.275$, and the trustworthy joint incentive $E$ for technology-based SMEs be 0.0, 7.5, 15.0, 30.0, 45.0, 55.0 and 70.0, respectively. When other parameters remain unchanged, the influence of the change of $E$ on the evolution path of the system is shown in Fig 12. The horizontal axis in Fig 12 indicates the probability $x$ of technology-based SMEs choosing the performance strategy, and the vertical axis indicates the probability $y$ of banks choosing the loan strategy.

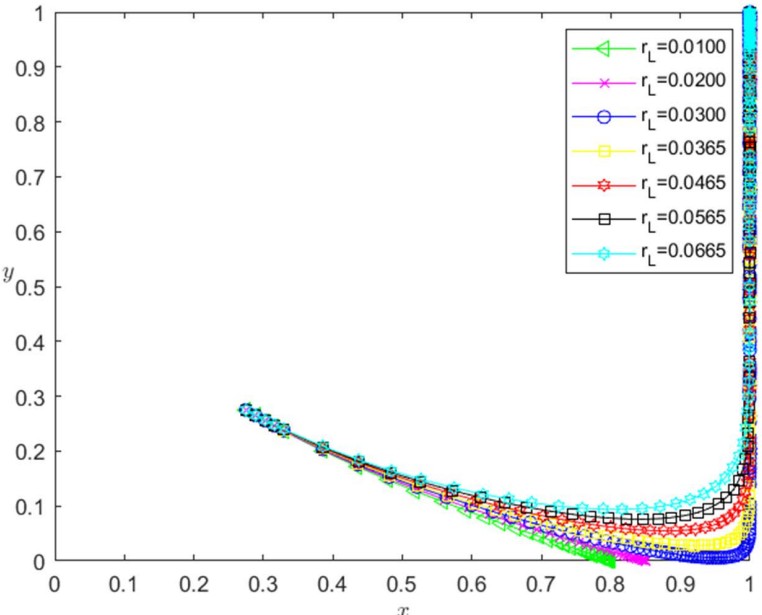

**Fig 10. The influence of the change of $r_L$ on the evolution path of the system.**

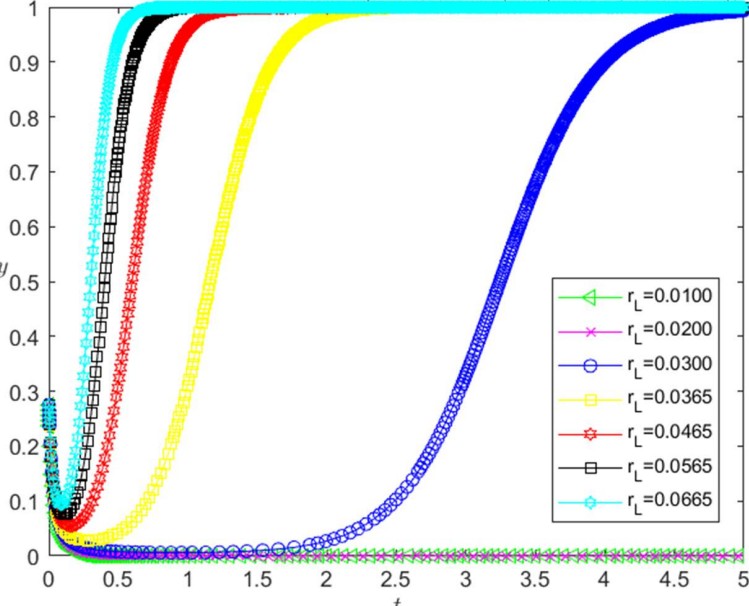

**Fig 11. The impact of the change of $r_L$ on the banks' strategy choice.**

The curve in Fig 12 shows the evolution path of the system under different $E$ values. It can be seen from Fig 12 that with the increase of $E$, the convergence speed of the system tends to Pareto optimal state is accelerated. And when $E < 15$, the system does not converge to the Pareto optimal state (1,1); when $E \geq 15$, the system converges to the Pareto optimal state (1,1).

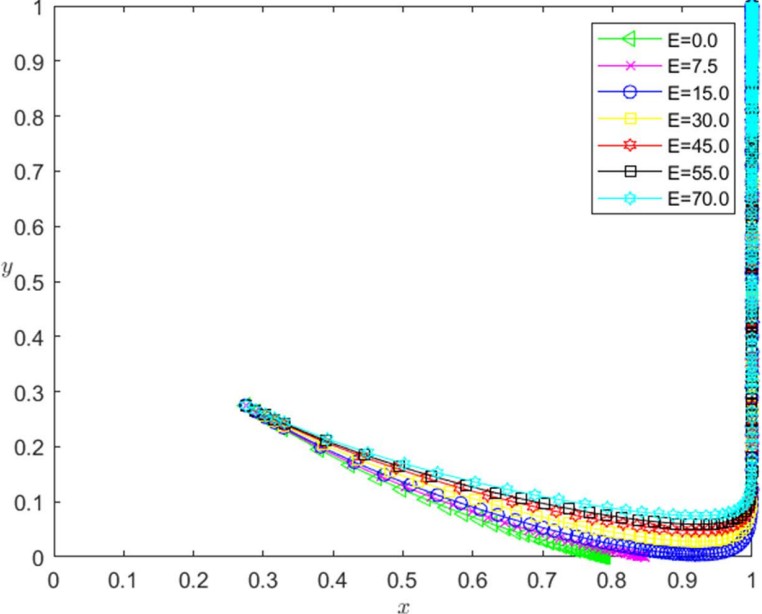

**Fig 12. The influence of the change of *E* on the evolution path of the system.**

Fig 13 reflects the impact of the change of *E* on the strategic choice of technology-based SMEs. The horizontal axis in Fig 13 represents the number *t* of simulation steps, and the vertical axis indicates the probability *x* of technology-based SMEs choosing the performance strategy. The curve in Fig 13 shows the evolution path of *x* under different *E* values. As

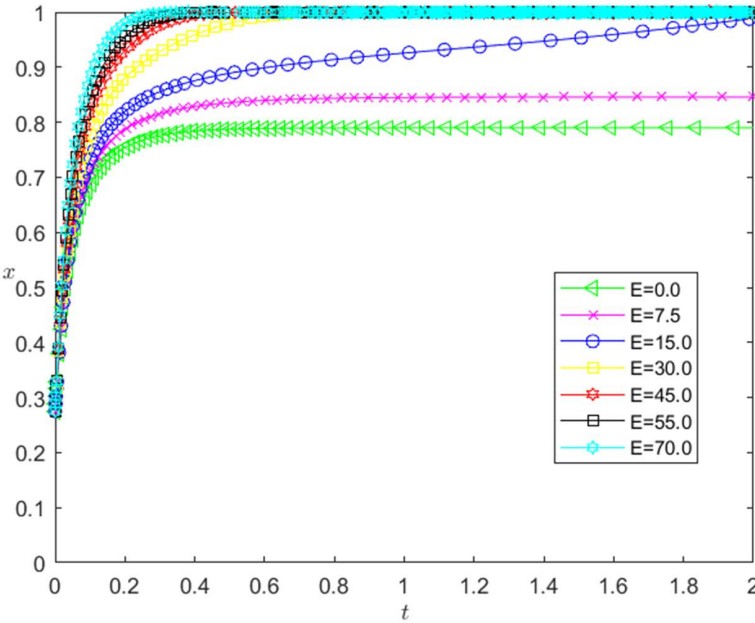

**Fig 13. The impact of the change of *E* on the technology-based SMEs' strategy choice.**

can be seen from Fig 13, the larger the *E* is, the faster the convergence rate of *x* approaching 1 is. And when *E* < 15, *x* converges below 0.9; when *E* ≥ 15, *x* converges to 1. This shows that the change of *E* has a positive impact on the evolution path of *x*. The larger the *E* is, the faster the technology-based SMEs choosing the performance strategy is. And the influence of *E* change on the evolution path of *x* has a positive threshold effect. After *E* is higher than a certain threshold, the technology-based SMEs will choose the performance strategy.

Fig 14 reflects the impact of the change of *E* on the bank's strategy choice. The horizontal axis in Fig 14 represents the number *t* of simulation steps, and the vertical axis indicates the probability *y* of banks choosing the loan strategy. The curve in Fig 14 shows the evolution path of *y* under different *E* values. As can be seen from Fig 14, the larger the *E* is, the faster the convergence rate of *y* approaching 1 is. And when *E* < 15, *y* converges to 0; when *E* ≥ 15, *y* converges to 1. This shows that the change of *E* has a positive impact on the evolution path of *y*. The greater the *E* is, the faster the banks choosing the loan strategy is. There is a positive threshold effect on the influence of *E* change on the evolution path of *y*. When *E* is higher than a certain threshold, banks will choose the loan strategy.

(6) The impact of F change on the strategic choice of two participants.

Let the initial probability $x_0 = 0.275$, $y_0 = 0.275$, and the dishonesty joint punishment *F* for technology-based SMEs be 55, 70, 85, 100, 115, 130 and 145. When other parameters remain unchanged, the influence of the change of *F* on the evolution path of the system is shown in Fig 15. The horizontal axis in Fig 15 indicates the probability *x* of technology-based SMEs choosing the performance strategy, and the vertical axis indicates the probability *y* of banks choosing the loan strategy. The curve in Fig 15 shows the evolution path of the system under different *F* values. It can be seen from Fig 15 that with the increase of *F*, the convergence speed of the system tends to Pareto optimal state is accelerated. And when *F* < 85, the system does not converge to the Pareto optimal state (1,1); when *F* ≥ 85, the system converges to the Pareto optimal state (1,1).

Fig 16 reflects the impact of the change of *F* on the strategic choice of technology-based SMEs. The horizontal axis in Fig 16 represents the number *t* of simulation steps, and the vertical axis indicates the probability *x* of technology-based

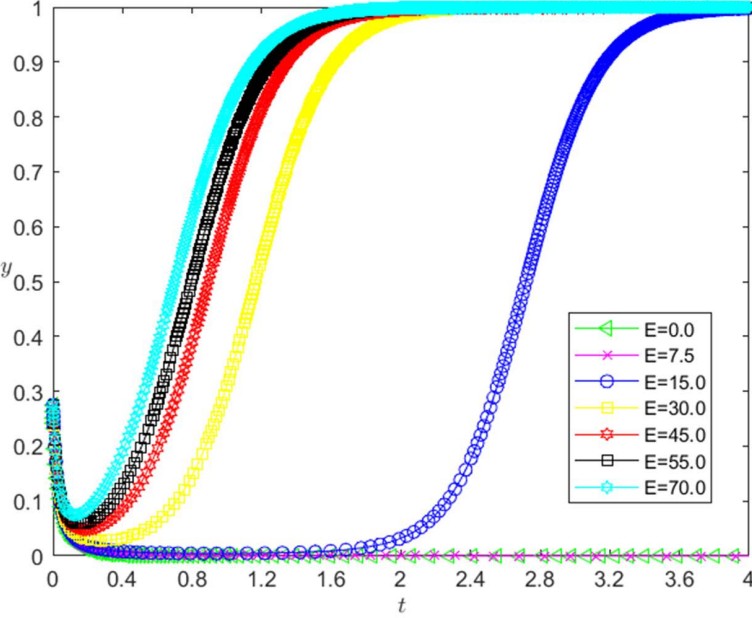

**Fig 14. The impact of the change of *E* on the banks' strategy choice.**

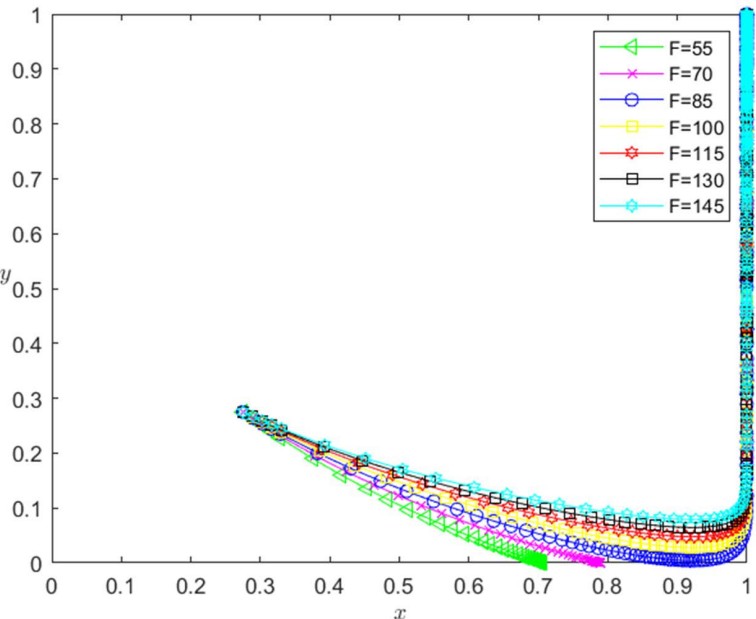

**Fig 15. The influence of the change of *F* on the evolution path of the system.**

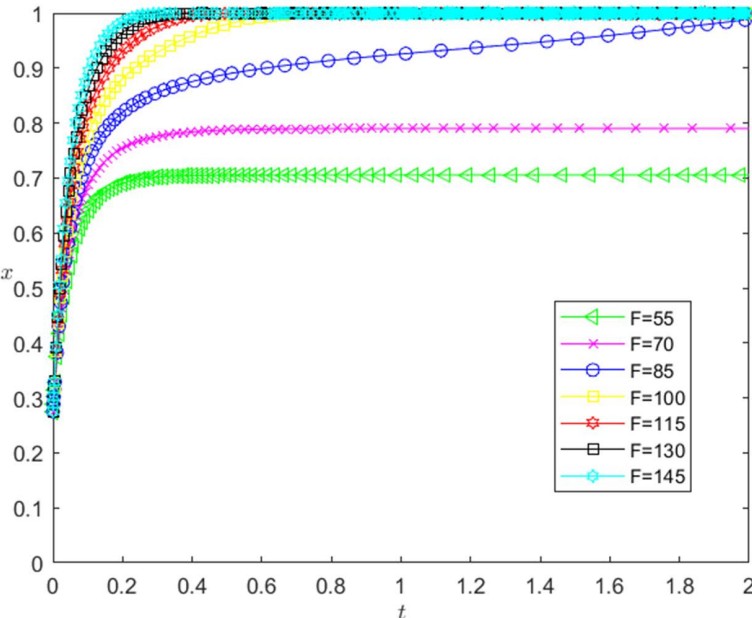

**Fig 16. The impact of the change of *F* on the technology-based SMEs' strategic choice.**

SMEs choosing the performance strategy. The curve in Fig 16 shows the evolution path of *x* under different *F* values. As can be seen from Fig 16, the larger the *F* is, the faster the convergence rate of *x* approaching 1 is. And when $F < 85$, *x* converges below 0.8; when $F \geq 85$, *x* converges to 1, which indicates that the change of *F* has a positive impact on the

evolution path of *x*. The larger the *F* is, the faster the technology-based SMEs choosing the performance strategy is. And the influence of *F* change on the evolution path of *x* has a positive threshold effect. After *F* is higher than a certain threshold, the technology-based SMEs will choose the performance strategy.

Fig 17 reflects the impact of the change of *F* on the bank's strategy choice. The horizontal axis in Fig 17 represents the number *t* of simulation steps, and the vertical axis indicates the probability *y* of banks choosing the loan strategy. The curve in Fig 17 shows the evolution path of *y* under different *F* values. As can be seen from Fig 17, the larger the *F* is, the faster the convergence rate of *y* approaching 1 is. And when $F < 85$, *y* converges to 0; when $F \geq 85$, *y* converges to 1. This shows that the change of *F* has a positive impact on the evolution path of *y*. The larger the *F* is, the faster the banks choosing the loan strategy is. And the influence of *F* change on the evolution path of *y* has a positive threshold effect. After *F* is higher than a certain threshold, the banks will choose the loan strategy.

In summary, the higher the initial probability of the two participants' strategy selection is, the faster the Pareto optimal state is reached. And the initial probability of the two participants' strategy selection needs to reach 27.5%, the system can achieve the Pareto optimal state (performance, loan). In addition, the influence of six key parameter changes on the strategic choice of two participants is as follows: 1) The changes of *k*, *E* and *F* have a positive impact on the evolution path of *x*, and they all have a positive threshold effect; the changes of $V_0$ and $\omega$ have a negative impact on the evolution path of *x*, and they all have a reverse threshold effect. In addition, the change of $r_L$ has no significant impact on the strategic choice of technology-based SMEs. 2) The changes of $r_L$, *E* and *F* have a positive impact on the evolution path of *y*, and they all have a positive threshold effect; the change of $V_0$ has a negative impact on the evolution path of *y*, and there is a reverse threshold effect. The change of $\omega$ has an inverted U-shaped influence on the evolution path of *y*, and there is a reverse threshold effect. In addition, the change of *k* has no significant impact on the bank's strategy choice.

## 4. Discussions

In the previous section of this paper, the numerical simulation results of the influence of six key parameter changes on the strategy selection of two participants are given, and the cause analysis is as follows:

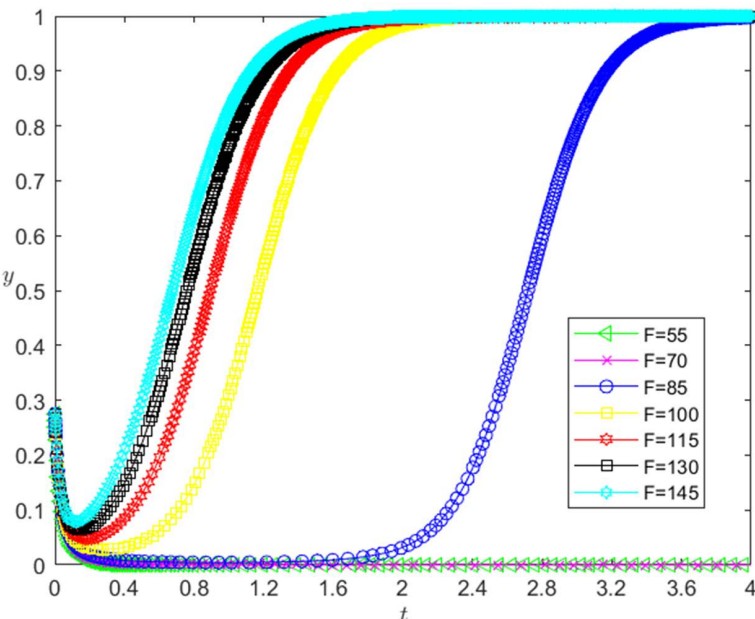

**Fig 17. The impact of the change of *F* on the banks' strategy choice.**

(1) The change of $k$ has a positive impact on the evolution path of $x$, and there is a positive threshold effect. The reason is that the greater the $k$ is, the greater the willingness of the enterprises to perform is, and the faster the enterprises choosing the performance strategy is. And $k$ needs to be large to a certain extent, so that the enterprises' willingness to perform has enough changes, the technology-based SMEs will choose the performance strategy. In addition, since $k$ is a decision variable for technology-based SMEs, not a decision variable for banks, $k$ change has no significant impact on banks' strategy choice.

(2) The change of $V_0$ has a negative impact on the evolution path of $x$, and there is a reverse threshold effect. The reason is that the smaller the $V_0$ is, the smaller the $\omega V_0(1 + r_L)$ is, and the larger the $k$ is (Note: $V_T$ is fixed at the end of the period), the greater the willingness of enterprises to perform is, and the faster the enterprises choosing the performance strategy is. And $V_0$ needs to be small to a certain extent, so that the enterprises' willingness to perform has enough changes, the technology-based SMEs will choose the performance strategy.

The change of $V_0$ has a negative impact on the evolution path of $y$, and there is a reverse threshold effect. The reason is that according to the bank's expected credit loss calculation formula: expected credit loss (EL) = default probability (PD)×default loss rate (LGD)×default risk exposure (EAD), the smaller the $V_0$ is, the smaller the loan principal $\omega V_0$ is, and thus the smaller the expected credit loss (EL) is, the faster the banks choosing the loan strategy is. And $V_0$ needs to be small to a certain extent, so that the bank's expected credit loss has enough changes, the banks will choose the loan strategy.

(3) The change of $\omega$ has a negative impact on the evolution path of $x$, and there is a reverse threshold effect. The reason is that the smaller the $\omega$ is, the smaller the $\omega V_0(1 + r_L)$ is, so that the larger the $k$ is (Note: $V_T$ is established at the end of the period), the greater the willingness of the enterprises to perform is, the faster the enterprises choosing the performance strategy is. And $\omega$ needs to be small to a certain extent, so that the enterprises' willingness to perform has enough changes, the technology-based SMEs will choose to perform the strategy.

The change of $\omega$ has an inverted U-shaped effect on the evolution path of $y$, and there is a reverse threshold effect. The reason is that, on the one hand, the greater $\omega$ is, the greater $\omega V_0$ is, the higher the expected loan interest income of the banks is, but the greater the expected credit loss of the banks is; on the other hand, the smaller $\omega$ is, the smaller $\omega V_0$ is, the smaller the expected credit loss of the banks is, but the lower the expected loan interest income of the banks is. Therefore, the banks will weigh the risks and benefits to determine an optimal $\omega^*$. And $\omega$ needs to be small to a certain extent, so that the banks' expected credit loss is within the acceptable range, the banks will choose the loan strategy.

(4) The change of $r_L$ has a positive impact on the evolution path of $y$, and there is a positive threshold effect. The reason is that the greater the loan interest rate $r_L$ is, the higher the expected loan interest income of the banks is, the faster the banks choosing the loan strategy is. And $r_L$ needs to be large to a certain extent, so that the banks' expected loan interest income has enough change, the banks will choose the loan strategy. In addition, because the value of $r_L$ is relatively small, $r_L$ has little effect on $k$, so it has little effect on the willingness of enterprises to perform, resulting in no significant impact of $r_L$ change on the strategic choice of technology-based SMEs.

(5) The change of $E$ has a positive impact on the evolution path of $x$, and there is a positive threshold effect. The reason is that the greater the $E$ is, the greater the willingness of technology-based SMEs to fulfill the contract is, the faster the technology-based SMEs choosing the performance strategy is. And the $E$ needs to be large to a certain extent, so that the willingness of the technology-based SMEs to fulfill the contract has enough changes, and the technology-based SMEs will choose the performance strategy.

The change of $E$ has a positive impact on the evolution path of $y$, and there is a positive threshold effect. The reason is that the greater the $E$, the smaller the estimated probability of default (PD) of the bank, the smaller the expected credit

loss (EL) of the banks is, and the faster the banks choosing the loan strategy is. And the $E$ needs to be large to a certain extent, so that the banks' expected credit loss has enough change, the banks will choose the loan strategy.

(6) The change of $F$ has a positive impact on the evolution path of $x$, and there is a positive threshold effect. The reason is that the greater the $F$ is, the smaller the willingness to default of the technology-based SMEs is, the greater the willingness to fulfill the contract is, the faster the technology-based SMEs choosing the performance strategy is. And $F$ needs to be large to a certain extent, so that the willingness of the technology-based SMEs to fulfill the contract has enough changes, and the technology-based SMEs will choose the performance strategy.

The change of $F$ has a positive impact on the evolution path of $y$, and there is a positive threshold effect. The reason is that the greater the $F$ is, the smaller the estimated probability of default (PD) of the banks is, the smaller the expected credit loss (EL) of the banks is, and the faster the banks choosing the loan strategy is. And $F$ needs to be large to a certain extent, so that the banks' expected credit loss has enough change, the banks will choose the loan strategy.

## 5. Conclusions

The possible marginal contributions of this paper are as follows: 1) Based on the perspective of enterprises' willingness to fulfill the contract, this paper defined the end-of-period value conversion coefficient of pledged property (EVCC) to measure the comparative relationship between the end-of-period value of the pledged intellectual property and the sum of principal and interest of the loan, and introduced it into the game payment matrix, an evolutionary game model of intellectual property pledge financing between technology-based SMEs and banks based on the EVCC was constructed. The results of asymptotic stability analysis show that when the condition ① is satisfied, the local equilibrium point $E_4 = (1, 1)$ is the asymptotic stability point of the replication dynamic system, that is, the strategy combination (performance, loan) is the evolutionary stability strategy (ESS). 2) The numerical simulation shows that the change of the EVCC $k$ has a positive impact on the evolution path of $x$, and there is a positive threshold effect. Among the parameters related to $k$, the change of the initial value $V_0$ of pledged intellectual property has a negative impact on the evolution path of $x$, and there is a reverse threshold effect, as well as the intellectual property pledge rate $\omega$; however, the change of loan interest rate $r_L$ has no significant impact on the strategic choice of technology-based SMEs. In addition, the change of $k$ has no significant impact on the banks' strategy choice. The change of $V_0$ has a negative impact on the evolution path of $y$, and there is a reverse threshold effect. The change of $\omega$ has an inverted U-shaped effect on the evolution path of $y$, and there is a reverse threshold effect. The change of $r_L$ has a positive impact on the evolution path of $y$, and there is a positive threshold effect. In addition, the changes of trustworthy joint incentive $E$ and dishonest joint punishment $F$ for technology-based SMEs not only have a positive impact on the evolution path of $x$, but also have a positive impact on the evolution path of $y$, and they all have a positive threshold effect.

The possible features of this paper are as follows:

1. The characteristics of the evolutionary game model

1) Compared with the evolutionary game model of intellectual property pledge financing between enterprises and banks constructed in [4–6,11,12,14–20,22], this paper introduces the EVCC $k$ in the evolutionary game model. Therefore, the influence of the comparative relationship between the end-of-period value of the pledged intellectual property and the sum of the loan principal and interest on the enterprises' willingness to fulfill the contract and thus on the enterprises' strategic choice is considered, which provides a new parameter for the evolutionary game analysis of the intellectual property pledge financing between the enterprises and the banks.

2) Compared with [12,17–19,22], which only consider the impact of the trustworthy joint incentive for enterprises or the dishonesty joint punishment for enterprises on the willingness to perform of enterprises in the evolutionary game model,

this paper considers the impact of the comparative relationship between the end-of-period value of pledged intellectual property and the sum of principal and interest of loan on enterprises performance willingness by introducing the EVCC $k$, thus broadening and deepening people's understanding of the factors affecting enterprises performance willingness.

3) Compared with the depreciation rate of patent rights proposed in [14], the EVCC $k$ defined in this paper not only reflects the depreciation of intellectual property at the end of the period ($k < 1/\omega(1 + r_L)$) but also reflects the appreciation of intellectual property at the end of the period ($k > 1/\omega(1 + r_L)$). It not only reflects the change of the end-of-period value of the pledged intellectual property but also reflects the comparative relationship between the end-of-period value of the pledged intellectual property and the sum of principal and interest of the loan; therefore, the connotation of $k$ is richer and the function is more diverse.

2. Characteristics of numerical simulation results

1) The higher the initial probability of strategy selection of technology-based SMEs and banks is, the faster the Pareto optimal state is reached. And the initial probability of the two sides' strategy selection needs to reach 27.5%, the system can achieve the Pareto optimal state of (performance, loan). Compared with the existing literatures, this paper draws similar conclusions.

2) The change of the EVCC $k$ has a positive impact on the speed of technology-based SMEs to choose the performance strategy, and there is a positive threshold effect. Compared with the existing literatures, this paper draws a new conclusion. Among the parameters related to $k$, the change of the initial value $V_0$ of pledged intellectual property has a negative impact on the speed of technology-based SMEs to choose the performance strategy, and there is a reverse threshold effect, as well as the change of intellectual property pledge rate $\omega$; compared with [17], this paper draws the opposite conclusion. The change of loan interest rate $r_L$ has no significant impact on the strategic choice of technology-based SMEs; compared with the existing literatures, this paper draws a new conclusion.

3) The change of trustworthy joint incentive $E$ has a positive impact on the speed of technology-based SMEs to choose the performance strategy, and there is a positive threshold effect, which is like the conclusion of [13,18,19]. The change of dishonesty joint punishment $F$ has a positive impact on the speed of technology-based SMEs to choose the performance strategy, and there is a positive threshold effect, which is like the conclusion of [13,17,22].

4) The change of the initial value $V_0$ of pledged intellectual property has a negative impact on the speed of banks to choose the loan strategy, and there is a reverse threshold effect. The change of intellectual property pledge rate $\omega$ has an inverted U-shaped impact on the speed of banks to choose the loan strategy, and there is a reverse threshold effect. Compared with [17], this paper draws the opposite conclusions. The change of loan interest rate $r_L$ has a positive impact on the speed of banks to choose the loan strategy, and there is a positive threshold effect. This is like the conclusion of [17]. The changes of trustworthy joint incentive $E$ and dishonest joint punishment $F$ have a positive impact on the speed of banks to choose the loan strategy, and they all have a positive threshold effect. Compared with the existing literatures, this paper draws a new conclusion.

Based on the perspective of enterprises performance willingness, this paper defined the end-of-period value conversion coefficient of pledged property (EVCC) to measure the comparative relationship between the end-of-period value of the pledged intellectual property and the sum of principal and interest of the loan and introduced it into the game payment matrix. According to the replication dynamic equation, the local equilibrium points of the replication dynamic system were obtained. By using the Friedman method, the asymptotic stability of the local equilibrium points was judged, and the evolutionary stability strategy (ESS) under certain conditions was obtained. Therefore, an evolutionary game model of intellectual property pledge financing between technology-based SMEs and banks based on the EVCC is constructed. The model in this paper enriches the multi-agent game theory framework of intellectual property pledge financing and has

high theoretical value. In addition, this paper analyzes the influence of the initial probability change of the strategic choice of technology-based SMEs and banks on the evolution path of the system, and the influence of the changes of the EVCC and other key parameters on the strategic choice of the two participants. The numerical simulation results of this paper can provide a decision-making reference for technology-based SMEs and banks to formulate intellectual property pledge financing strategies and have high application value.

The policy implications of this paper are as follows: 1) In the decision-making of intellectual property pledge rate and loan interest rate, banks should appropriately consider the impact of intellectual property pledge rate and loan interest rate on the willingness of enterprises to perform through loan principal and interest. 2) Banks (or asset appraisal institutions) should use big data, artificial intelligence and other technologies to establish a dynamic evaluation system for the market value of pledged intellectual property rights, to realize real-time tracking of the market value of pledged intellectual property rights. 3) According to the dynamic change of the relationship between the market value of pledged intellectual property rights and the sum of principal and interest of loans, banks should establish an early warning model for the willingness of enterprises to perform and incorporate it into the enterprises credit risk early warning system.

This paper constructs an evolutionary game model of intellectual property pledge financing between technology-based SMEs and banks based on the EVCC. However, market practice shows that there may be multiple subjects such as enterprises, banks, third-party intermediaries (or platforms), and governments in intellectual property pledge financing. Therefore, future research can consider including game subjects such as governments, and third-party intermediaries (or platforms) in this model. The model in this paper considers the comparative relationship between the end-of-period value of pledged intellectual property and the sum of loan principal and interest, the impact of joint incentive of enterprises trustworthiness, and joint punishment of enterprises dishonesty on enterprises performance willingness. Future research can further consider the impact of enterprises ownership structure, enterprises governance structure, and other factors on enterprises performance willingness. In addition, 12 parameters are assigned in the numerical simulation. However, for the parameters that are difficult to quantify, such as the joint incentive of enterprises trustworthiness and the joint punishment of enterprises dishonesty, this paper can only be assigned subjectively, which has strong subjectivity. Future research can consider using a fuzzy multi-attribute decision-making method to quantify the above parameters.

## Author contributions

**Conceptualization:** Mu Zhang.

**Data curation:** Li-na Dong, Mu Zhang.

**Investigation:** Li-na Dong, Mu Zhang.

**Methodology:** Mu Zhang.

**Software:** Li-na Dong.

**Supervision:** Mu Zhang.

**Validation:** Li-na Dong.

**Writing – original draft:** Li-na Dong, Mu Zhang.

**Writing – review & editing:** Li-na Dong, Mu Zhang.

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
