## [Decision Letter · Decision Letter 0]

28 May 2025

Thank you for submitting your manuscript to PLOS ONE. After careful consideration, we feel that it has merit but does not fully meet PLOS ONE’s publication criteria as it currently stands. Therefore, we invite you to submit a revised version of the manuscript that addresses the points raised during the review process.

**ACADEMIC EDITOR: **

Please address the comments below from the three reviewers along with the academic editor.

We look forward to receiving your revised manuscript.

Kind regards,

Ahmed Eltweri, Ph.D

Academic Editor

PLOS ONE

 [This research was funded by the Research Project of Humanities and Social Sciences of Colleges and Universities in Guizhou, grant number 2024RW96, “Research on the problem of ‘financing difficulties’ and ‘expensive financing’ of small and medium-sized enterprises in Guizhou”.]. 

Additional Editor Comments:

This paper presents a novel and methodologically sound evolutionary game model to analyse the strategic interactions between technology-based SMEs and banks in the context of intellectual property pledge financing. A key innovation is the introduction of the “end-of-period value conversion coefficient”, which attempts to model the willingness to perform financial obligations.

The theoretical framework is robust, and simulations are thorough. However, some key areas including parameter transparency, empirical contextualisation, presentation clarity, and data/code availability, require attention to meet PLOS ONE’s criteria for scientific validity, reporting transparency, and accessibility. Below are the principal areas where revisions are necessary to elevate the manuscript to a publishable standard:

The current title is too long and unwieldy, reducing accessibility and search engine optimisation. I suggest shortening and clarify the title in line with the 2nd reviewer recommendation

The manuscript uses subjectively assigned parameter values (e.g., penalty costs, return rates) without adequate empirical or theoretical backing. The author must consider add a Table or Appendix justifying parameter choices. As well as consider sensitivity ranges where values are hard to justify precisely.

The manuscript lacks information about data/code availability, which is required by PLOS ONE for transparency and reproducibility. The authors should upload the MATLAB code used for simulations to a repository.

The model is purely theoretical and simulation-based. Hence, some conclusions imply real-world policy implications without empirical backing. Therefore, the authors must clearly state in the Discussion that conclusions are theoretical and require empirical validation. As well as suggesting future work to calibrate the model using real data from Chinese IP pledge financing programs or banks.

Several critical equations are unnumbered or embedded in long, dense paragraphs. Therefore, authors must number all equations for ease of reference. And add brief intuitive explanations or summaries after each key equation set. Or perhaps consider using figures or diagrams to represent the game tree or model structure.

The figure captions lack detail, and variable meanings are not consistently clarified. Authors must ensure all axis labels are complete. In addition, expand captions to interpret the figures.

While the literature review is substantial, it could benefit from more recent international literature on IP financing or trust-based SME financing strategies. I suggest that author must incorporate newer references from 2022 onwards. Include Fintech and SME trust modelling, IP collateral valuation challenges and Game-theoretic applications in finance.

The manuscript refers to practical relevance but lacks concrete recommendations. In addition add a dedicated section (or at least a paragraph in the Conclusion) with practical implications which should include threshold values for policy intervention; How banks or governments might use the coefficient “k” and Incentive design for SME trustworthiness

Finally, the English is mostly clear, but some technical and mathematical sections are difficult to follow, shorter sentences and more active voice, rephrase dense mathematical paragraphs with bullets or boxed summaries where possible.

Reviewers' comments:

Reviewer's Responses to Questions

**Comments to the Author**

1. Is the manuscript technically sound, and do the data support the conclusions?

Reviewer #1: Yes

Reviewer #2: Yes

Reviewer #3: Yes

2. Has the statistical analysis been performed appropriately and rigorously?

Reviewer #1: Yes

Reviewer #2: Yes

Reviewer #3: Yes

3. Have the authors made all data underlying the findings in their manuscript fully available?

Reviewer #1: Yes

Reviewer #2: Yes

Reviewer #3: Yes

4. Is the manuscript presented in an intelligible fashion and written in standard English?

Reviewer #1: Yes

Reviewer #2: Yes

Reviewer #3: No

Reviewer #1: Your manuscript is a well-structured and technically rigorous study on IP pledge financing using evolutionary game theory. It is suitable for publication after minor revisions, particularly:

Strengthening the justification for parameter assignments.

Providing a data availability statement.

Improving readability in mathematical sections

Reviewer #2: 1. Title Clarity

Suggested revision: The title is long and dense. Consider rephrasing slightly for clarity and international accessibility.

Proposed title: Evolutionary Game Model for Intellectual Property Pledge Financing: Evidence from End-of-Period Value Conversion in SME-Bank Interactions

2. Abstract: The abstract lacks numerical context for key parameters and simulation results.

Suggested revision: Add specific numerical outcomes from the simulations (e.g. thresholds for the coefficient where ESS is achieved) to support claims.

Example: “When the value conversion coefficient exceeds 1.15, cooperation becomes evolutionarily stable...”

3. Terminology Clarification

The term "end-of-period value conversion coefficient" is introduced early but not clearly explained until later.

Suggested revision: Define this term succinctly and mathematically in the abstract or early in the introduction, ideally with an equation or variable symbol (e.g. λ) for consistency.

4. Literature Review

The literature section does not include recent empirical work on IP pledge valuation mechanisms.

Suggested revision: Add 1–2 references from the last 2–3 years focusing on financial valuation of IP in emerging markets or fintech applications in SME lending.

5. Figure Annotations

Figures lack detailed captions and axis labels are not always intuitive (e.g. "repayment willingness" as a Y-axis).

Suggested revision: Include full mathematical labels (e.g., "Probability of strategy adoption: Repayment (p)") and clearly describe what the curves represent in the caption.

6. Numerical Simulation Parameters

Parameter values for simulation (e.g., payoff matrix elements, initial population proportions) are buried in the text.

Suggested revision: Add a dedicated table summarising all input parameters for the numerical simulation, with justifications or citations if applicable.

7. Grammar and Style

Several sentences contain grammatical errors or awkward phrasing.

Examples & Fixes:

“To improve the market efficiency of intellectual property pledge financing...” → “This study aims to improve the efficiency of IP pledge financing...”

“...construct the end-of-period value conversion coefficient of pledged property to measure the comparative relationship...” → “...define a conversion coefficient to measure the ratio between pledged IP value at maturity and total loan repayment.”

8. Policy Implications Section

The policy section is too generic.

Suggested revision: Include specific recommendations, such as “Development of national IP valuation guidelines” or “Incentives for banks offering IP-backed financing.”

9. Mathematical Consistency

Equation numbers are inconsistent and sometimes missing.

Suggested revision: Ensure all key equations are numbered and referenced in the text for easy traceability.

Reviewer #3: In the second line of the first paragraph, SMEs are not specifically explained.

The article uses past and past perfect tenses, which may indicate machine translation.

The references include Chinese literature, and I suggest changing them to English.

**Do you want your identity to be public for this peer review?** For information about this choice, including consent withdrawal, please see our Privacy Policy

Reviewer #1: No

Reviewer #2: No

Reviewer #3: No

---

## [Author Response · Author response to Decision Letter 1]

5 Aug 2025

Dear Editors and Reviewers:

Thank you for your letter and for the reviewers’ comments concerning our manuscript entitled “Evolutionary Game Model of Intellectual Property Pledge Financing Between Technology-based SMEs and Banks Based on the End-of-period Value Conversion Coefficient of Pledged Property” (ID: PONE-D-24-56353). Those comments are all valuable and very helpful for revising and improving our paper, as well as the important guiding significance to our research. We have studied comments carefully and have made correction which we hope meet with approval. Revised portions are marked in red in the paper. The main corrections in the paper and the responds to the reviewer’s comments are as follows:

Responds to the reviewer’s comments:

Reviewer #1:

1. Response to comment: (Strengthening the justification for parameter assignments.)

Response: Considering the Reviewer’s suggestion, we have added specific reference sources for each parameter assignment. See Section 3.1, paragraph 2 for details.

2. Response to comment: (Providing a data availability statement.)

Response: We have added a Data availability statement at the end of this paper according to the Reviewer’s suggestion. See page 28 for details.

3. Response to comment: (Improving readability in mathematical sections.)

Response: Considering the Reviewer’s suggestion, we have carefully modified the mathematical expressions in the mathematical sections. See Section 2.2, Section 2.3, Section 2.5 and Section 2.6 for details.

Special thanks to you for your good comments.

Reviewer #2:

1. Response to comment: (Title Clarity: Suggested revision: The title is long and dense. Consider rephrasing slightly for clarity and international accessibility. Proposed title: Evolutionary Game Model for Intellectual Property Pledge Financing: Evidence from End-of-Period Value Conversion in SME-Bank Interactions.)

Response: Considering the Reviewer’s suggestion, we have revised the manuscript title to: Evolutionary Game Model of Intellectual Property Pledge Financing Between Technology-based SMEs and Banks Based on the EVCC. See page 1 for details.

2. Response to comment: (Abstract: The abstract lacks numerical context for key parameters and simulation results. Suggested revision: Add specific numerical outcomes from the simulations (e.g. thresholds for the coefficient where ESS is achieved) to support claims. Example: “When the value conversion coefficient exceeds 1.15, cooperation becomes evolutionarily stable...”.)

Response: Considering the Reviewer’s suggestion, we have supplemented the threshold values of key parameters in the abstract. See Abstract for details.

3. Response to comment: (Terminology Clarification: The term "end-of-period value conversion coefficient" is introduced early but not clearly explained until later. Suggested revision: Define this term succinctly and mathematically in the abstract or early in the introduction, ideally with an equation or variable symbol (e.g. λ) for consistency.)

Response: According to the Reviewer’s suggestion, we have modified the title of Section 2.2 to: Definition of EVCC, and proposed Definition 1, using the symbol k to represent the end-of-period value conversion coefficient of pledged property (EVCC). See Section 2.2 for details.

4. Response to comment: (Literature Review: The literature section does not include recent empirical work on IP pledge valuation mechanisms. Suggested revision: Add 1–2 references from the last 2–3 years focusing on financial valuation of IP in emerging markets or fintech applications in SME lending.)

Response: Considering the Reviewer’s suggestion, we have added two new references: [20] and [23].

[20] Chen, Y., Yuan, J. L., Ren, K. J., et al. (2024). Analysis of the Evolution Game of Intellectual Property Pledge Financing from the Perspective of Blockchain. Brand and standardization, (05), 167-170.

[23] Ran, C. J., Zhang, Y. R., & Huang, W. J. (2025). Does artificial intelligence affect the credit risk of intellectual property pledge financing? - Evolutionary Game Analysis of Credit Risk of Intellectual Property Pledge Financing Based on Game Theory. Library construction, (03), 36-47+59. doi: https://doi.org/10.19764/j.cnki.tsgjs.20250614

5. Response to comment: (Figure Annotations: Figures lack detailed captions and axis labels are not always intuitive (e.g. "repayment willingness" as a Y-axis). Suggested revision: Include full mathematical labels (e.g., "Probability of strategy adoption: Repayment (p)") and clearly describe what the curves represent in the caption.)

Response: According to the Reviewer’s suggestion, we have added a detailed description of the horizontal axis and the vertical axis for each figure in the text, and explained the meaning of the curve in the figure. See Section 3.2.1 and Section 3.2.2 for details.

6. Response to comment: (Numerical Simulation Parameters: Parameter values for simulation (e.g., payoff matrix elements, initial population proportions) are buried in the text. Suggested revision: Add a dedicated table summarizing all input parameters for the numerical simulation, with justifications or citations if applicable.)

Response: Considering the Reviewer’s suggestion, we have summarized all parameter assignments in Table 4, and added specific reference sources for each parameter assignment. See Section 3.1 for details.

7. Response to comment: (Grammar and Style: Several sentences contain grammatical errors or awkward phrasing. Examples & Fixes: “To improve the market efficiency of intellectual property pledge financing...” → “This study aims to improve the efficiency of IP pledge financing...”; “...construct the end-of-period value conversion coefficient of pledged property to measure the comparative relationship...” → “...define a conversion coefficient to measure the ratio between pledged IP value at maturity and total loan repayment.”)

Response: According to the Reviewer’s suggestion, we have carefully corrected the grammar errors in both the abstract and the main text. The sections with significant changes include: Abstract, Section 1, Section 2.2, Section 2.3, Section 2.5, Section 2.6, Section 3.2.1, Section 3.2.2, Section 4, and Section 5.

8. Response to comment: (Policy Implications Section: The policy section is too generic. Suggested revision: Include specific recommendations, such as “Development of national IP valuation guidelines” or “Incentives for banks offering IP-backed financing.”)

Response: Considering the Reviewer’s suggestion, we have added a natural paragraph in Section 5 to express policy recommendations. See Section 5, penultimate paragraph.

9. Response to comment: (Mathematical Consistency: Equation numbers are inconsistent and sometimes missing. Suggested revision: Ensure all key equations are numbered and referenced in the text for easy traceability.)

Response: Considering the Reviewer’s suggestion, we have put the unnumbered formula in Section 2.2 into the text, to maintain the continuity and consistency of the formula number. See Section 2.2 for details.

Special thanks to you for your good comments.

Reviewer #3:

1. Response to comment: (In the second line of the first paragraph, SMEs are not specifically explained.)

Response: Considering the Reviewer’s suggestion, we have supplemented the definition of technology-based SMEs in Section 1, paragraph 1. See Section 1, paragraph 1 for details.

2. Response to comment: (The article uses past and past perfect tenses, which may indicate machine translation.)

Response: According to the Reviewer’s suggestion, we have carefully corrected the grammar errors in both the abstract and the main text. The sections with significant changes include: Abstract, Section 1, Section 2.2, Section 2.3, Section 2.5, Section 2.6, Section 3.2.1, Section 3.2.2, Section 4, and Section 5.

3. Response to comment: (The references include Chinese literature, and I suggest changing them to English.)

Response: Considering the Reviewer’s suggestion, we have carefully proofread all references to ensure that all references are in English.

Special thanks to you for your good comments.

Other changes:

1. We have removed the previously weakly related reference [16], and added the reference [33] in the parameter assignment.

2. Due to the addition of 3 references and deletion of 1 reference, we have adjusted the reference numbers.

3. We have added a “Funding State” and a “Conflicts of Interest” at the end of this paper.

We tried our best to improve the manuscript and made some changes in the manuscript. These changes will not influence the content and framework of the paper. And here we did not list the changes but marked in red in revised paper.

We appreciate for Editors/Reviewers’ warm work earnestly, and hope that the correction will meet with approval.

Once again, thank you very much for your comments and suggestions.

---

## [Editor Report · Decision Letter 1]

15 Aug 2025

Evolutionary Game Model of Intellectual Property Pledge Financing Between Technology-based SMEs and Banks Based on the EVCC

PONE-D-24-56353R1

Dear Dr. Zhang,

We’re pleased to inform you that your manuscript has been judged scientifically suitable for publication and will be formally accepted for publication once it meets all outstanding technical requirements.

Kind regards,

Ahmed Eltweri, Ph.D

Academic Editor

PLOS ONE

Additional Editor Comments (optional):

Dear Dr. Zhang,

Thank you for submitting the revised version of your manuscript entitled "Evolutionary Game Model of Intellectual Property Pledge Financing Between Technology-based SMEs and Banks Based on the EVCC" (ID: PONE-D-24-56353R1) to PLOS ONE.

I have reviewed your responses to the reviewers’ comments and the revised manuscript in detail. The revisions have addressed all substantive concerns raised in the initial review, including clarification of the EVCC definition, incorporation of numerical thresholds into the abstract, improvements to mathematical clarity, addition of recent literature, clearer figure annotations, inclusion of a parameter summary table, enhancement of the policy implications section, and corrections to language and formatting.

The manuscript is now clear, complete, and meets the journal’s publication criteria. I am pleased to inform you that your article is accepted for publication in PLOS ONE in its current form.

Our production team will be in touch with you regarding proofs and final publication details. Please ensure that all final files, including figures and supplementary materials, are ready for transfer to production.

On behalf of the editorial board, I thank you for choosing PLOS ONE as the outlet for your work and look forward to seeing your contribution published.

Congratulations on this achievement.

Kind regards,

Dr Ahmed Eltweri

Academic Editor

PLOS ONE
---

## [Editor Report · Acceptance letter]

PONE-D-24-56353R1

PLOS ONE

Dear Dr. Zhang,

I'm pleased to inform you that your manuscript has been deemed suitable for publication in PLOS ONE. Congratulations! Your manuscript is now being handed over to our production team.

Kind regards,

on behalf of

Dr. Ahmed Eltweri

Academic Editor

PLOS ONE